# Nuclear chromosome locations dictate segregation error frequencies

Sjoerd J. Klaasen[1], My Anh Truong[2,6], Richard H. van Jaarsveld[1,6], Isabella Koprivec[3], Valentina Štimac[3], Sippe G. de Vries[2], Patrik Risteski[3], Snježana Kodba[3], Kruno Vukušić[3], Kim L. de Luca[1], Joana F. Marques[1], Elianne M. Gerrits[1], Bjorn Bakker[4], Floris Foijer[4], Jop Kind[1,5], Iva M. Tolić[3], Susanne M. A. Lens[2] & Geert J. P. L. Kops[1✉]

Chromosome segregation errors during cell divisions generate aneuploidies and micronuclei, which can undergo extensive chromosomal rearrangements such as chromothripsis[1–5]. Selective pressures then shape distinct aneuploidy and rearrangement patterns—for example, in cancer[6,7]—but it is unknown whether initial biases in segregation errors and micronucleation exist for particular chromosomes. Using single-cell DNA sequencing[8] after an error-prone mitosis in untransformed, diploid cell lines and organoids, we show that chromosomes have different segregation error frequencies that result in non-random aneuploidy landscapes. Isolation and sequencing of single micronuclei from these cells showed that mis-segregating chromosomes frequently also preferentially become entrapped in micronuclei. A similar bias was found in naturally occurring micronuclei of two cancer cell lines. We find that segregation error frequencies of individual chromosomes correlate with their location in the interphase nucleus, and show that this is highest for peripheral chromosomes behind spindle poles. Randomization of chromosome positions, Cas9-mediated live tracking and forced repositioning of individual chromosomes showed that a greater distance from the nuclear centre directly increases the propensity to mis-segregate. Accordingly, chromothripsis in cancer genomes[9] and aneuploidies in early development[10] occur more frequently for larger chromosomes, which are preferentially located near the nuclear periphery. Our findings reveal a direct link between nuclear chromosome positions, segregation error frequencies and micronucleus content, with implications for our understanding of tumour genome evolution and the origins of specific aneuploidies during development.

Aneuploidy, a state in which the genome content of cells deviates from an integer multiple of the haploid set, can cause miscarriages and developmental syndromes and is strongly associated with tumourigenesis[7,11–13]. Aneuploidies typically result from chromosome segregation errors during cell divisions. Such errors can be caused by, for example, altered microtubule dynamics, hyperstable kinetochore–microtubule attachments, cohesion problems or weakened spindle assembly checkpoint (SAC) activity[2,14–17]. Mitotic errors also lead to the formation of micronuclei, which are prone to nuclear envelope rupture leading to extensive genomic rearrangements[3–5,18]. Cancers exhibit distinct aneuploidy and rearrangement patterns—for instance, when originating from different organs, after colonizing distant sites or after relapse—suggesting convergent evolution towards an optimal karyotype under distinct selective pressures[11,19–23]. Aneuploidies found in preimplantation or aborted embryos are likewise not random[10,24]. Although selection probably plays a crucial role, it is unknown whether the process

that causes aneuploidy is biased and may create a non-random genomic substrate for selection. In support of this possibility, severe perturbations of the spindle or centromere can affect segregation of some chromosomes more than others[25–27].

## Chromosome segregation error frequencies

To examine the propensity of individual chromosomes for mis-segregation, we used single-cell karyotype sequencing (scKaryo-seq[8]) to assess hundreds of karyotypes from diploid cells that underwent an aberrant mitosis. scKaryo-seq allows for high fidelity and high-throughput determination of the copy number state of all chromosomes in single cells. RPE1-hTERT cells synchronized in G2 (Extended Data Fig. 1a) were allowed to proceed to mitosis in the presence or absence of a low concentration (62.5 nM) of Cpd-5, a small-molecule inhibitor of the mitotic kinase MPS1 (ref. [28]). Low Cpd-5

[1]Oncode Institute, Hubrecht Institute-KNAW (Royal Academy of Arts and Sciences) and University Medical Centre Utrecht, Utrecht, the Netherlands. [2]Oncode Institute, Centre for Molecular Medicine, University Medical Centre Utrecht, Utrecht University, Utrecht, The Netherlands. [3]Ruđer Bošković Institute, Zagreb, Croatia. [4]Department of Ageing Biology/ERIBA, University of Groningen, University Medical Centre Groningen, Groningen, the Netherlands. [5]Department of Molecular Biology, Faculty of Science, Radboud Institute for Molecular Life Sciences, Radboud University Nijmegen, Nijmegen, The Netherlands. [6]These authors contributed equally: My Anh Truong, Richard H. van Jaarsveld. ✉e-mail: g.kops@hubrecht.eu

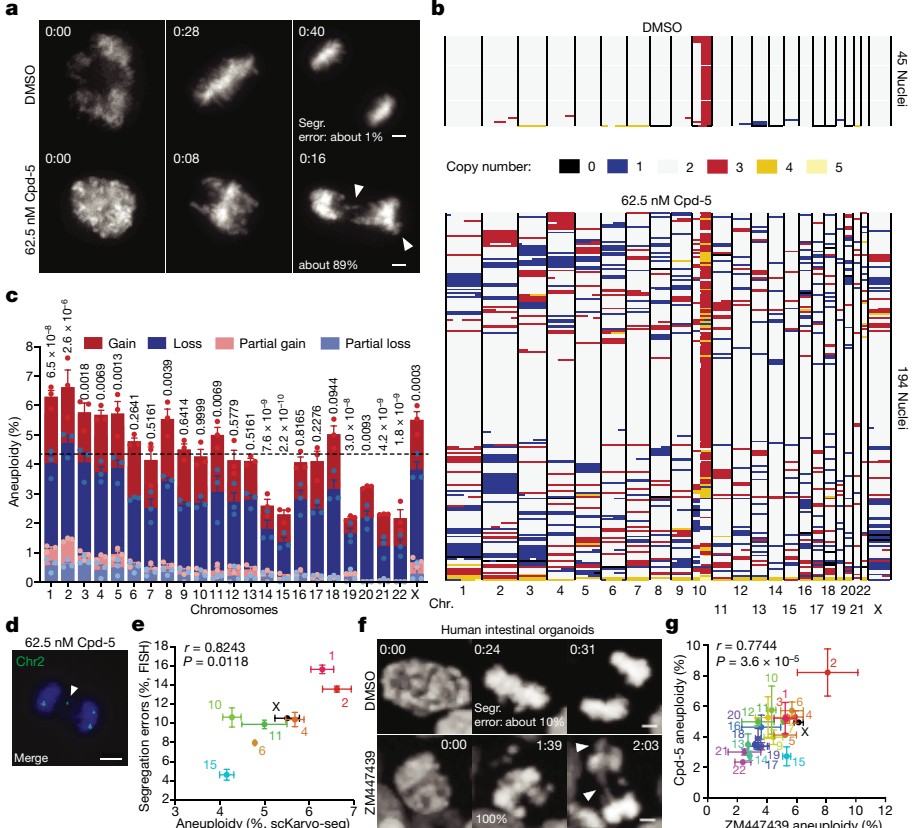

**Fig. 1 | Segregation error frequencies of chromosomes during error-prone mitosis are non-random. a**, Representative still images of synchronous RPE1-hTERT H2B-mNeon cells undergoing mitosis in the presence of Cpd-5. Arrowheads indicate mis-segregating chromosomes (scale bars, 5 μm). **b,c**, Representative replication (**b**) and quantification (**c**) showing the copy number states of synchronised RPE1-hTERT cells treated with DMSO or Cpd-5. Each row represents one nucleus, and different colours represent copy number states. The graph shows aneuploidy percentages per chromosome for three independent experiments (mean ± s.e.m., two-tailed binomial test, *n* = 410 aneuploid cells). Different colours denote whether the chromosome has been lost, gained or partially lost or gained. The horizontal black line at 4.34% depicts the expected random chance of mis-segregation of each chromosome. Numbers at the top of columns in **c** are *P* values. **d,e**, Representative FISH image (**d**) and quantification (**e**) of RPE1-hTERT cells as treated in **b** but fixed 45 min

after release from RO-3306 (scale bar, 5 μm). Arrowhead indicates a lagging chromosome 2. The graph plots the percentage of chromosomes positive for corresponding FISH probes on the *y* axis versus the aneuploidy percentages determined in **c** (mean ± s.e.m., two-tailed Pearson's correlation coefficient; chr1, *n* = 327; chr2, *n* = 307; chr4, *n* = 311; chr6, *n* = 290; chr10, *n* = 322; chr11, *n* = 313; chr15, *n* = 286; chrX, *n* = 314). Three independent experiments were performed. **f**, Representative still images of human intestinal organoids treated with either DMSO as a control or the Aurora B inhibitor ZM447439 (scale bars, 5 μm). Arrowheads indicate mis-segregating chromosomes. **g**, Plot comparing aneuploidy percentages of human intestinal organoids after overnight Cpd-5 versus ZM447439 treatment (mean ± s.e.m., two-tailed Pearson's correlation coefficient, *n* = 217 and *n* = 64 aneuploid cells, respectively). The experiment was performed in duplicate.

compromises attachment error correction and the SAC response (Extended Data Fig. 1b–d), leading to nearly all cells undergoing anaphase with a few misaligned and lagging chromosomes (Fig. 1a, Extended Data Fig. 1e,f and Supplementary Videos 1 and 2). These are among the most common mitotic errors observed in cancer cells[2,8,29]. Our synchronization-and-release procedure did not affect short-term viability (Extended Data Fig. 2a,b). Individual nuclei isolated 4 h following error-prone mitosis were subjected to scKaryo-seq. Low Cpd-5 had caused an average of 5.5 segregation error events per aneuploid cell, totalling 2,417 quantifiable whole and structural chromosome copy number alterations (Fig. 1b). Strikingly, aneuploidy landscapes revealed that segregation error probabilities were not equal among chromosomes: chromosomes 1–5, 8, 11 and X had a significantly higher probability of mis-segregation than expected from a random error frequency of 4.3% (Bonferroni correction, *P* = 0.0022). Chromosomes 14, 15 and 19–22, on the other hand, had significantly lower probabilities (Fig. 1c). A similarly biased distribution of aneuploidies was seen in diploid BJ-hTERT fibroblasts and human intestinal organoids, and in a near-tetraploid RPE1-hTERT cell line (Extended Data Fig. 2c–g and

Extended Data Fig. 3a–f), showing that a mis-segregation bias occurs in different cell types and culture conditions. Fluorescence in situ hybridization (FISH) for centromeres of chromosomes 1, 2, 4, 6, 10, 11, 15 and X on anaphase figures further verified that the non-random aneuploidy landscapes observed by scKaryo-seq were due to biased segregation errors in anaphase (Fig. 1d,e). Importantly, similar segregation error probabilities were seen when mitotic errors were induced by other means. First, small-molecule inhibition of Aurora B, although inducing severe errors in monolayer cultures[30,31], caused mild segregation errors in human intestinal organoids, allowing isolation of daughter nuclei (Supplementary Video 3). scKaryo-seq of these nuclei showed a segregation error bias highly similar to that of Cpd-5-treated cells (Fig. 1f,g). Second, a low concentration of nocodazole perturbs microtubule assembly rates, a proposed cause of chromosomal instability in colorectal cancer cells[15]. Low nocodazole caused misaligned chromosomes in RPE1-hTERT cells that subsequently frequently mis-segregated in anaphase[32] (Extended Data Fig. 3g,h and Supplementary Video 4). Although it was not possible to perform scKaryo-seq due to the rarity of segregation errors in this condition,

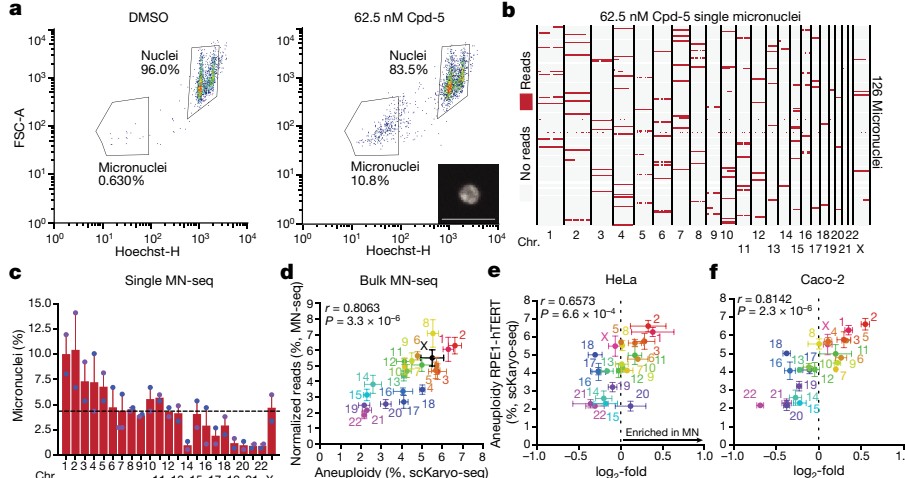

**Fig. 2 | Micronuclear content reflects segregation error bias.**
**a**, Representative flow cytometry plots of RPE1-hTERT nuclei and micronuclei. Hoechst height (Hoechst-H) signal is plotted against forward scatter area (FSC-A). Image depicts micronucleus imaged after sorting (scale bar, 5 μm). **b,c**, Representative MN-seq data (**b**) and quantification (**c**) of single micronuclei isolated from cells treated with Cpd-5 (mean ± s.e.m., two independent experiments, $n$ = 222 micronuclei). Each line in **b**. represents a single micronucleus; red denotes the presence of a chromosomal region.
**d**, Plot comparing bulk MN-seq data of around 8,000 micronuclei, shown as the percentage of reads mapped to a certain chromosome normalised for chromosome size and the aneuploidy percentages determined in Fig. 1c (mean ± s.e.m., two-tailed Pearson's correlation coefficient). Four independent experiments were performed. **e,f**, Plots comparing $\log_2$-fold enrichment of micronuclear DNA determined from bulk sequencing of two chromosomally unstable cancer cell lines (Hela (**e**) and Caco-2 (**f**)) versus the aneuploidy percentages from Fig. 1c (mean ± s.e.m., two-tailed Pearson's correlation coefficient, $n$ = around 6,000 MN). Three independent experiments were performed for each cell line.

centromere FISH for six chromosomes (1, 6, 7, 15, 18 and X) showed that misalignment frequencies were significantly different between chromosomes and followed a similar trend to Cpd-5/AurBi-induced aneuploidy landscapes (Extended Data Fig. 3i,j). Taken together, we conclude that chromosome segregation errors of various origins in non-transformed cells are non-random.

## Chromosome micronucleation frequencies

Erroneously segregated chromosomes frequently become encapsulated in micronuclei. Chromosomes in micronuclei can undergo extensive rearrangements such as chromothripsis, which plays an important role in tumour evolution[3–5,18]. Because we observed a bias in initial aneuploidy landscapes after segregation errors, we examined whether this would translate into a biased micronuclear content. We therefore adapted a fluorescent activated cell sorter (FACS)-based approach[33] to isolate micronuclei and assess their content by sequencing, a protocol we refer to as MN-seq. MN-seq performed on single- or bulk-sorted micronuclei from Cpd-5-treated cells revealed a striking non-random content that correlated strongly with the pattern of mis-segregation frequencies of chromosomes (Fig. 2a–d and Extended Data Fig. 4a–c). This was verified by FISH (Extended Data Fig. 4d,e). A similarly biased micronuclear content was observed for cells treated with a low dose of nocodazole (Extended Data Fig. 4f–h). We next examined whether micronuclear content is non-random in naturally occurring micronuclei of cancer cells. MN-seq of five chromosomally unstable cancer cell lines showed that non-random micronuclear content existed in all lines, and that two lines (Caco-2 and HeLa) showed a bias highly similar to that of Cpd-5/AurBi-treated diploid cells (Fig. 2e,f and Extended Data Fig. 4i–k). Of note, recent FISH analysis showed a similar micronuclear content bias in glioblastoma cells[34]. These data indicate that chromosomes with a higher mis-segregation probability more frequently become entrapped in micronuclei. Because naturally occurring micronuclei most probably arise from relatively recent mis-segregation events, our data further suggest that segregation errors in various cancer cells are non-random and may be caused by similar underlying mechanisms.

## Origin of non-random segregation errors

Having established that frequencies of mis-segregation and entrapment in micronuclei are not equal for all chromosomes, we next aimed to understand the origins of this bias. We therefore compared segregation error frequencies with chromosome characteristics such as centromere size or the ratio of $p$- to $q$-arm lengths, but did not find any correlation (Extended Data Fig. 5a,b). Chromosome size and gene density, on the other hand, did correlate (directly and inversely, respectively) with segregation error frequencies (Extended Data Fig. 5c,d). Chromosome size is nonetheless unlikely to be a direct cause for the observed bias, because the two X chromosomes that differ in size by 24.4% in RPE1-hTERT cells, owing to one being fused to a $q$-arm of chromosome 10, had identical mis-segregation frequencies (Extended Data Fig. 5e). Moreover, some chromosomes of near-identical size had profoundly different mis-segregation frequencies (for example, 13 versus 14, 15 versus 16 or 18 versus 19) (Fig. 1c). We next considered the locations of chromosomes in interphase nuclei, because they correlate directly with chromosome size and inversely with gene density[35–37] but can differ between chromosomes of near-identical size. In support of this, four out of five acrocentric chromosomes (14, 15, 21 and 22), which reside near the centrally located nucleoli, had low mis-segregation frequencies (Fig. 1c). Segregation error frequencies of chromosomes in RPE1-hTERT cells strongly correlated with the densities of lamina-associated chromosome domains (LADs) (Fig. 3a), and with published distances of chromosomes from the centre of the nucleus[37] (Extended Data Fig. 5f). Chromosome size correlated to LAD densities in RPE1-hTERT and other cells, both diploid (TIG3-hTERT and hESCs) and aneuploid (HeLa, U2OS, K562 and HT1080), showing that in a wide array of cell lines, larger chromosomes reside in the nuclear periphery more frequently than smaller ones[38–42] (Extended Data Fig. 5g–m). Centromere FISH of six chromosomes (1, 2, 6, 15, 17 and 18) further verified this in both non-transformed and transformed cell lines (Fig. 3b,c and Extended Data Fig. 6a–e).

To further explore the proposal that chromosome location can cause a segregation error bias, we live-imaged chromosome movement during mitosis in cells with labelled centromeres and centrosomes (Fig. 3d,

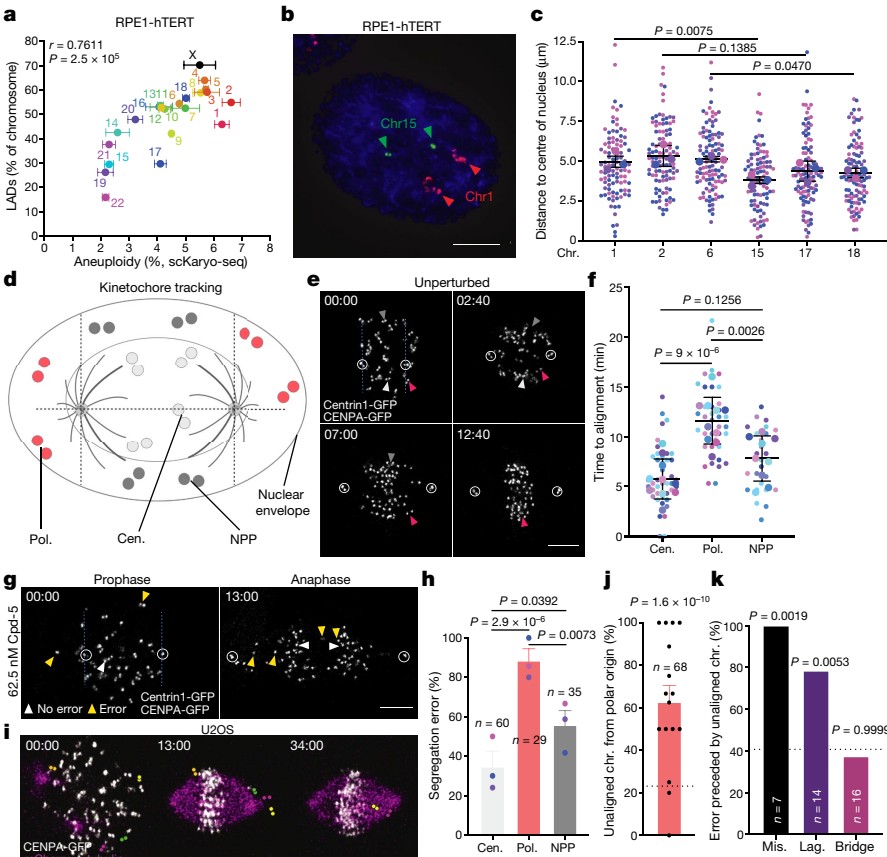

**Fig. 3 | Segregation error probabilities correlate with nuclear chromosome locations. a**, Plot comparing aneuploidy percentages after Cpd-5 treatment as in Fig. 1c, and the percentage of LADs for every chromosome in RPE1-hTERT cells (mean + s.e.m., two-tailed Pearson's correlation coefficient, *n* = 2 independent DamID experiments). **b,c**, Representative FISH image of RPE1-hTERT cells (**b**) and quantification of the distance of chromosomes from the centre of the nucleus (**c**) (mean ± s.d., two-tailed ratio *t*-test, *n* = 99, 106, 108, 98, 106 or 107 chromosomes, respectively; scale bar, 5 μm). The experiment was performed in triplicate. **d**, Schematic depicting the nucleus of a cell with kinetochore pairs in different-coloured circles depending on their location relative to the two centrosomes. **e,f**, Representative still images (**e**) and quantification (**f**) of the alignment time of kinetochores in RPE1-hTERT CENPA-GFP/Centrin1-GFP cells (scale bar, 5 μm). Arrowheads indicate specific kinetochore pairs, circles centrosomes. The experiment was performed ten

times (mean ± s.d., one-way analysis of variance with Tukey's post hoc test, *n* = 38, 29 and 35 chromosome pairs, respectively). **g,h**, As in **e,f**, respectively, but instead cells were treated with 62.5 nM Cpd-5 and mis-segregations were quantified (**h**). Arrowheads denote kinetochore pairs (white, no error; yellow, error); scale bar, 5 μm. The experiment was performed in triplicate (mean ± s.e.m., two-tailed Fisher's exact test, *n* = 43, 35 and 26 chromosomes). **i–k**, Representative still images (**i**) and quantification (**j,k**) of kinetochore behaviour in U2OS cells expressing CENPA-GFP/mCherry-α-tubulin (scale bar, 5 μm). Dashed lines represent the expected percentage of unaligned chromosomes (**j**) or errors (**k**), which are based on the number of polar kinetochore pairs at the start of nuclear envelope breakdown (NEBD) and the number of unaligned chromosomes at the start of metaphase. The experiment was performed 16 times (mean ± s.e.m., two-tailed Fisher's exact test). Mis., misaligned; lag., laggard.

Extended Data Fig. 6f and Supplementary Videos 5 and 6) and analysed the trajectories of three categories of chromosome: centrally located (cen.), peripheral (located behind a spindle pole (pol.)) and peripheral (located between spindle pole planes (NPP)) (Fig. 3d). This showed that peripheral chromosomes required significantly longer to congress than central ones under unperturbed conditions, and more readily mis-segregated as misaligned or lagging chromosomes upon treatment with Cpd-5 (Fig. 3e–h). Among peripheral chromosomes, the polar ones congressed more slowly and mis-segregated more frequently than non-polar ones. This was true also when comparing only polar and non-polar peripheral chromosomes with an equal starting distance from the metaphase plate (Extended Data Fig. 6f–k). We observed a similar phenomenon in chromosomally unstable cancer cells in which peripheral, polar chromosomes had a threefold higher probability of being misaligned than expected on the basis of the frequency of polars (Fig. 3i,j and Supplementary Video 7). Moreover, misalignments and laggards in anaphase were observed significantly more frequently in cells with such misaligned metaphase chromosomes than in cells without (18/77 versus 3/112, respectively; Fig. 3k), suggesting that

mis-segregating chromosomes may often arise from misaligned chromosomes. These observations suggest that the peripheral location of chromosomes—and, even more so, how this location relates to spindle poles—delays biorientation and congression and thereby contributes to the high frequency of mis-segregation of peripheral chromosomes.

## The role of nuclear chromosome location

We next aimed to examine whether the distinct locations of chromosomes in an interphase nucleus are directly responsible for variation in segregation error probability. In a first approach, we randomized the location of chromosomes in mitosis before inducting erroneous anaphases. This was done by a short treatment with monastrol, which allows for microtubule-based chromosome movements in a monopolar spindle and simultaneously renders most if not all chromosomes essentially spindle pole proximal (Fig. 4a,b). scKaryo-seq of cells released from monastrol into low Cpd-5 showed that the probability of chromosome mis-segregation was substantially altered in two different cell lines (Fig. 4c and Extended Data Fig. 7a–e). In a second

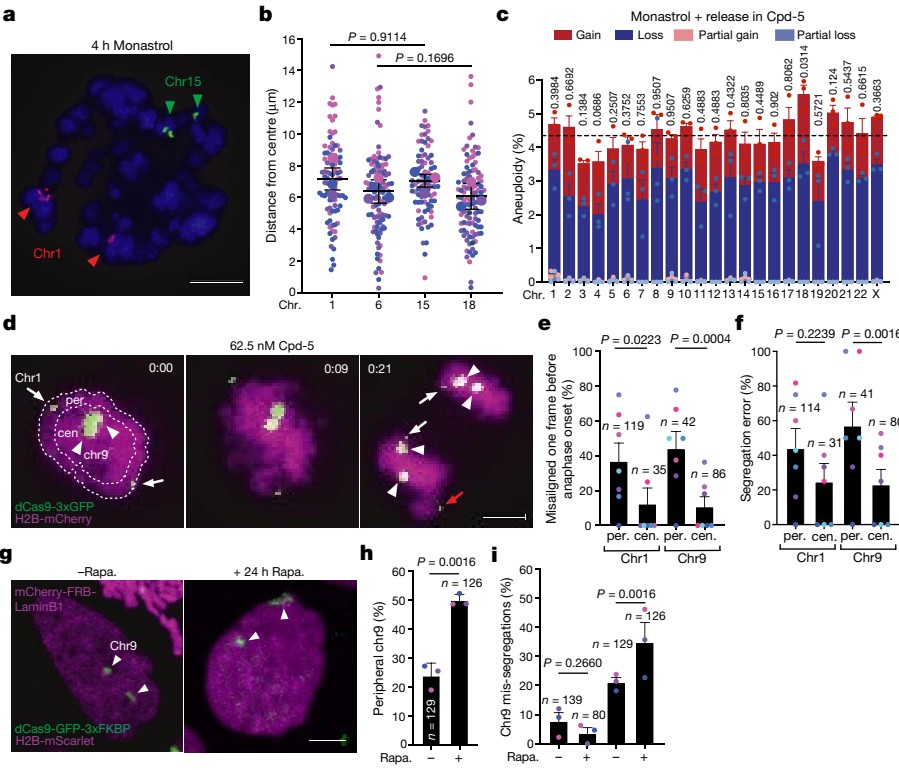

**Fig. 4 | Distance of chromosomes from the nuclear centre dictates mis-segregation probability. a,b**, Representative FISH images (**a**) and quantification (**b**) of the distance of FISH probes from the centre in RPE1-hTERT cells synchronised as before and treated for 4 h with monastrol (mean ± s.d., two-tailed ratio *t*-test, *n* = 89, 110, 87 and 109 chromosomes, respectively). Data were pooled from three independent experiments (scale bar, 5 µm). **c**, Quantification of aneuploidy numbers determined by scKaryo-seq of RPE1-hTERT cells treated as in **i**, followed by release in Cpd-5, shake-off and replating. Data were pooled from three independent experiments (mean ± s.e.m., two-tailed binomial test, *n* = 373 aneuploid cells). **d–f**, Still images (**d**) and quantification of central (cen.) versus peripheral (per.) chromosomes 1 (arrows) and 9 (arrowheads) misaligned immediately before anaphase onset (**e**) and subsequently mis-segregating in RPE1-hTERT cells (**f**) (scale bar, 5 µm). White arrows indicate properly segregating chromosomes, while the red arrow follows a mis-segregating one. The experiment was performed in septuplicate (mean ± s.d., two-tailed Fisher's exact test). **g–i**, Still images (**g**) and quantification of the peripherality (**h**) of chromosome 9 in DLD1-expressing dCas9-GFP-3xFKBP mCherry-FRB-LaminB1 after 24 h of rapalog (rapa.); scale bar, 5 µm. Cells were followed live to determine mis-segregations in Cpd-5 (**i**). The experiment was performed in triplicate (mean ± s.e.m., two-tailed Fisher's exact test).

approach, we leveraged the ability to image specific chromosomes using repeat-specific guide RNAs and fluorescently labelled dCas9 (refs. [27,43]). Chromosomes that are on average mostly peripheral can occasionally be found in the nuclear interior, and vice versa. This allowed us to directly relate the location of a specific chromosome to its behaviour in mitosis (Fig. 4d). Live imaging of chromosomes 1 (small foci) and 9 (large foci) verified that, before nuclear envelope breakdown, chromosome 1 was more often peripheral compared with chromosome 9 but was centrally located in around 20% of cells[27] (Extended Data Fig. 8a,b and Supplementary Video 8). Conversely, chromosome 9 could be found near the nuclear periphery in roughly 35% of cells. Regardless of identity, chromosomes starting from a peripheral location were significantly more often misaligned before anaphase onset in Cpd-5-treated cells compared with when they started from a more central position (threefold for chromosome 1, fourfold for chromosome 9) (Fig. 4e). Importantly, a peripheral starting position substantially increased mis-segregation rates (misalignments and laggards) compared with a central starting position (1.8- and 2.5-fold for chromosomes 1 and 9, respectively) (Fig. 4f). A similar result was obtained after tethering and subsequent tracking of LacI-GFP to chromosome 11-specific LacO repeat sites in fibrosarcoma HT1080 cells[44] (Extended Data Fig. 8c,d). In a final approach, we forced relocation of chromosome 9 from a central location to the periphery by conditionally tethering it to the nuclear lamina via rapalog-controlled dimerization of dCas9 and Lamin B1. Following 24 h of incubation with rapalog, the frequency by which

chromosome 9 was found near the periphery was increased nearly twofold (Fig. 4g,h) which, by itself, did not compromise its correct segregation during normal divisions (Fig. 4i). Upon error-prone divisions following the addition of Cpd-5, however, relocation of chromosome 9 resulted in a nearly twofold increase in its mis-segregation rate compared with non-relocated chromosome 9 (Fig. 4i). Taken together, these data led us to conclude that chromosome segregation error frequencies are directly influenced by the location in the interphase nucleus before the onset of mitosis.

## Discussion

Our data are consistent with a model in which the location of chromosomes in the nucleus dictates probabilities of segregation errors of individual chromosomes and their entrapment in micronuclei (Extended Data Fig. 9). We propose that peripheral chromosomes mis-segregate more often than central ones because they need to travel greater distances to reach the metaphase plate, more regularly undergo lateral or merotelic microtubule interactions and/or experience additional delays by being located behind spindle poles[45–49]. It may therefore be a common outcome of processes that cause chromosomal instability, consistent with occurence of biased mi-segregations when we disrupt the SAC, attachment error correction or microtubule dynamics. Additional biases from different underlying processes are nonetheless likely to exist[25–27], and could for example affect aneuploidies in human female meiosis[24].

Of note, chromosome size also correlates with aneuploidy landscapes after the first few error-prone mitotic divisions of human embryos[10]. In mice, errors during these divisions have been attributed to a weakened SAC[17]. The nuclear location of chromosomes may thus also explain non-random aneuploidies during embryonic development. Interestingly, a recent survey of chromothriptic events in 2,658 human cancers showed that their occurrence strongly correlates with chromosome size and thus with nuclear position[9] (Extended Data Fig. 10). Non-random segregation errors may therefore affecct the dynamics of recurring aneuploidy and genomic rearrangement patterns seen in cancer, and thereby influence tumour growth, metastasis and relapse. Tissue-specific chromosome locations[50] are likely to differentially affect these dynamics.

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

## Methods

### Cell culture

Cell lines RPE1-hTERT (Flp-In) (a gift from the laboratory of P. Jallepalli), Caco-2 (a gift from the laboratory of H. Clevers), HeLa (a gift from the laboratory of M. Vermeulen), HT-29 (a gift from the laboratory of H. Clevers), U2OS (a gift from the laboratory of S. Lens) and WiDr (a gift from the laboratory of H. Clevers) were cultured in DMEM/F12 and GlutaMAX supplement (Gibco), supplemented with 9% foetal bovine serum (FBS, Sigma-Aldrich) and 1% penicillin/streptomycin (Sigma-Aldrich). Cell lines U2OS DamID, U2OS CENPA and BJ-hTERT (a gift from the laboratory of R. Medema), HCT116 (a gift from the laboratory of H. Clevers) and HT1080 were cultured in DMEM, high-glucose GlutaMAX supplement and pyruvate (Gibco), supplemented with 10% FBS and 1% penicillin/streptomycin. Human small intestine duodenum and ileum organoids (a gift from the laboratory of H. Clevers) were cultured as described previously[51]. Rather than WNT conditioned medium, WNT surrogate was used (0.15 nM, U-Protein Express). DLD1 cells (a gift from D. Cimini) were cultured in RPMI and GlutaMAX supplement (Gibco), supplemented with 9% FBS and 50 μg ml$^{-1}$ penicillin/streptomycin. To generate RPE1-hTERT Flp-in H2B-mNeon cells, cells were transduced with a lentivirus containing an H2B-mNeon-IRES-puromycin construct. Selection was performed with 10 μg ml$^{-1}$ puromycin (Sigma-Aldrich) for 48 h. Organoids were transduced with the same construct without selection. Tetraploid RPE1-hTERT cells were generated by treatment of original RPE1-hTERT cells with 62.5 nM Cpd-5 for 48 h every 7 days for 4 weeks, after which tetraploid colonies grew out for an additional 22 weeks. Monoclonal RPE1-hTERT dCas9-3×GFP and DLD1 dCas9-GFP-3×FKBP FRB-mCherry-LaminB1 lines were generated by transduction with a dCas9-3×GFP or dCas9-GFP-3×FKBP lentivirus, followed by single-cell sorting. Next, FRB-mCherry-LaminB1 was lentivirally introduced in DLD1 cells. HT1080 cells containing a LacO-array in chromosome 11 (a gift from W. Bickmore) were transduced with LacI-GFP-FKBP and FRB-mCherry-LaminB1 and cloned by single-cell sorting. Cell lines were tested for mycoplasma contamination and not authenticated.

### scKaryo-seq

RPE1-hTERT Flp-in cells were plated in a six-well plate (Corning) at 40% confluency and treated with palbociclib (250 nM; Selleck Chemicals). After 24 h, cells were washed three times with warm medium and treated with RO-3306 (5 μM; Tocris Bioscience). After 16 h, cells were washed three times for 5 min at 37 °C with warm medium containing DMSO, Cpd-5 (62.5 nM; a gift from R. Medema) or monastrol (200 μM; Sigma-Aldrich). Cpd-5-treated cells were cultured for a further 4 h before harvesting. Monastrol-treated cells were washed three times with warm medium containing 62.5 nM Cpd-5. Mitotic cells were collected by shake-off and plated in a new well of a six-well plate for 4 h. BJ-hTERT cells were plated in a six-well plate at 40% confluency and treated with 31.25 nM Cpd-5 for 16 h. All cells were trypsinized and stored at −20 °C for further processing. Single G1 nuclei of RPE1-hTERT Flp-in cells or single nuclei of BJ-hTERT cells were sorted as described previously[8]. Human intestinal organoids were plated 1 day before treatment for 16 h with 5 μM ZM447439 (Selleck Chemicals) or 10 μM EdU (Thermofisher) for 3 h, washed three times for 5 min with warm medium, incubated with 62.5 nM Cpd-5 for 16 h and fixed using 70% ice-cold ethanol. Ethanol was removed by one wash with PBS, and cells were incubated for 10 min with the Click-iT reaction cocktail (Click-iT EdU proliferation assay). The reaction cocktail was washed away and replaced with a PBS/DAPI mix. Single G1 nuclei in the case of ZM447439 or EdU-positive G1 cells were sorted in 384-well plates. Tetraploid RPE1-hTERT cells were plated at 40% confluency and treated with 62.5 nM Cpd-5 for 24 h. G1 nuclei were sorted. HCT116 cells were synchronized for 16 h using monastrol, released and treated with Cpd-5 as described for RPE1-hTERT cells. Plates were stored at −20 °C.

NlaIII-based library preparation was performed as described previously, with several modifications[8]. Cell lysis was performed for 2 h at 55 °C with 8 mg ml$^{-1}$ Proteinase K (Fisher Scientific) in 1× CutSmart (New England Biolabs) and heat inactivation at 80 °C for 10 min. Adaptors were ligated with 100 nl of 100 nM barcoded, double-stranded NLAIII adaptors and 400 nl of 10 U T4 DNA ligase (New England Biolabs) in 1× T4 DNA ligase buffer (New England Biolabs), supplemented with 3 mM ATP (Invitrogen) at 16 °C overnight. Samples were sequenced on an Illumina NextSeq500 or 2000 at 1× 75 or 1× 100 base pairs (bp), respectively. After sequencing, mapping (bwa aln 0.7.12 and python 2.7.5) and Aneufinder (v.1.2.0) plotting and copy number variations of whole and partial chromosomes were determined manually. Chromosome 8 of human intestinal organoids was not quantified because this chromosome was heterogeneously aneuploid under the control condition.

### Centromere FISH

Cells were plated on 12-mm round glass coverslips (Superior Marienfeld). To validate scKaryo-seq segregation error bias, cells were synchronized and treated with Cpd-5 as described above. Cells were fixed 45 min after release from RO-3306, at −20 °C with 75% methanol and 25% acetic acid. To determine the distance of chromosomes from the centre of the nucleus, cells were plated 1 day before fixation. To determine nuclear chromosome territories of monastrol-treated mitotic cells, cells were synchronized as described above and incubated for 4 h in monastrol, then subsequently fixed. After fixation, coverslips were air-dried and incubated for 2 min with 2× saline-sodium citrate (SSC) at room temperature. Coverslips were washed in series with 70%, 85% and 100% ethanol and air-dried. Next, 1.2 μl of a red and green satellite enumeration probe (Cytocell) and 1.6 μl of hybridization solution per coverslip were spotted on a glass slide. Coverslips were placed upside down on the probe solution and incubated at 75 °C for 2 min. Coverslips were incubated at room temperature for 4–16 h, followed by 2 min incubation at 72 °C with 0.25× SSC (pH 7.0). Coverslips were washed for 30 s with 2× SSC 0.5% Tween-20 at room temperature, incubated with DAPI and mounted using ProLong Gold antifade (Molecular Probes).

Image acquisition was done on a DeltaVision RT system (Applied Precision/GE Healthcare) with a ×1.40/100 numerical aperture (NA) UplanSApo objective (Olympus) as z-stacks at 0.5 μm intervals. For deconvolution, SoftWorx (Applied Precision/GE Healthcare, v.6.5.2) was used. Image analysis and quantification was done using Fiji ImageJ (v.2.0.0).

FISH segregation error frequencies were determined by counting the number of mis-segregating FISH-positive chromosomes and dividing that by the total number of mis-segregating chromosomes.

Chromosomes in low-dose nocodazole were considered misaligned when FISH-positive chromosomes were physically separated from the metaphase plate; this number was then divided by the total number of FISH-positive chromosomes.

To measure the distance of chromosomes from the centre of the nucleus, we determined the centroid $X$ and $Y$ coordinates of the three different thresholded channels (DAPI, red probe and green probe). The centre of monastrol-treated cells was determined using a custom ImageJ script, which measures the centre of mass of thresholded DAPI particles.

### Live imaging

To time mitotic phases, RPE1-hTERT Flp-in H2B-mNeon cells were plated in a black, glass-bottom, 96-well plate (Corning) at 40% confluency and synchronized as described for scKaryo-seq. Cells were imaged on an Andor CSU-W1 spinning disk (50 μm disk) with a ×0.75/20 NA dry objective lens (Nikon). A 488 nm laser was used for sample excitation, with filters between 540 and 50 nm bandpass for emission. Images were acquired using an Andor iXon-888 EMCCD camera. Nine z-slices of 2 μm were imaged for 4 h every 1 min. NEBD was defined as one frame

before extensive chromosome movement. Images were acquired using NIS-elements (Nikon, v.5.30.04).

To determine the time from condensation to anaphase onset and segregation errors, we used a Nikon Ti-E motorized microscope equipped with a Zyla 4.2Mpx sCMOS camera (Andor) and a ×1.3/40 NA oil objective lens (Nikon). Fluorescence excitation was done using a Spectra X LED illumination system (Lumencor) and Chroma-ET filter sets. Nine z-slices of 2 μm were imaged every 4 min for 4 h. The same videos were also used to determine cell survival.

To examine cell survival for MN-seq, RPE1-hTERT Flp-in and BJ-hTERT cells were plated at 40% confluency. Cells were imaged on the same micro-scope used for determination of segregation errors. DIC and a ×0.45/10 NA objective lens (Nikon) were used to visualize cells every 3–5 min for 16 h.

Human intestinal organoids were imaged as described previously[8].

To determine mis-segregations in cells treated with low-dose nocodazole, RPE1-hTERT H2B-eYFP cells were plated at 40% confluency 1 day before imaging. Next, cells were treated with nocodazole (48 nM; Sigma-Aldrich) and imaged on an Andor CSU-W1 spinning disk (50 μm disk) with a ×1.45/100 NA oil objective lens (Nikon). A 488 nm laser was used for sample excitation and filters between 540 and 50 nm bandpass for emission. Images were acquired using an Andor iXon-888 EMCCD camera. Nine z-slices of 2 μm were imaged for 16 h every 3 min.

To compare the behaviour of polar and non-polar chromosomes, RPE1-hTERT cells stably expressing both CENPA-GFP and Centrin1-GFP (a gift from A. Khodjakov) were imaged on the Expert Line easy3D STED microscope system (Abberior Instruments) using Prairie View (5.4.64.500) and Inspector (Abberior Instruments, v.16.3) with 485 and 640 nm lasers using a ×60/1.2 UPLSAPO 60×W water objective (Olympus) and an avalanche photodiode (APD) detector. Low-dose (1:100,000) SPY-595-DNA was added to detect the moment of nuclear envelope breakdown, and low-dose (1:50,000) SPY-640-tubulin (Spirochrome, AG) was added to distinguish between poles and kine-tochores, as well as to enable pole tracking when the Centrin1 signal was not easily detectable in a specific frame. Six z-slices of 1 μm were taken every 20 s. Immediately after nuclear envelope breakdown, the edges of the nucleus were manually drawn to determine the relative nuclear position of tracked chromosomes by dividing the nucleus into three equally spaced concentric areas. Chromosomes were considered central if they resided in the two innermost shells or were touching the second-most outer ring. Positions of both centrosomes were also determined at that point. Each kinetochore pair was followed manually in a maximum-intensity projection. The positions and trajectories of the kinetochore pairs were additionally verified in single z-planes of a z-stack in Fiji (v.1.53f51/1.53s30/1.53r), as well as in Imaris 3D Viewer (v.9.8.0). One pair each of polar and non-polar peripheral chromo-somes with the same distance to the metaphase plate were selected from the same cell.

U2OS kinetochore tracking experiments were performed with a U2OS cell line stably expressing CENPA-GFP, mCherry-α-tubulin and photoactivatable-GFP-α-tubulin (a gift from M. Barisic and H. Maiato). Cells were imaged using a Bruker Opterra I multipoint scanning con-focal microscope system, as previously described[52]. Image acquisi-tion was performed at 1 min intervals with z-stacks of 15 slices at 1 μm spacing. Misaligned kinetochores included all pairs of kinetochores displaced from the metaphase plate in the frame when elongation of the prometaphase spindle reached its peak, which was defined as the final point at which the separation of two centrosomes showed a continuous increase in spindle length for two consecutive frames >1 μm. Spatial x and y coordinates of unaligned kinetochores were extracted in every time frame using the Low Light Tracking Tool (v.0.10), an ImageJ plugin, as previously described[53]. The tracking of kinetochores in x and y planes was performed on individual imaging z-planes. Around 10–15% of unaligned kinetochore pairs could not be successfully tracked in all frames, mainly owing to cell and spindle

movements in the z-direction over time. Spindle poles were manually tracked with points placed in the centre of the pole structure, in the z-plane in which the tubulin signal was highest. Aligned kinetochore pairs were manually tracked in two dimensions. All unaligned pairs in the NEBD frame were double-checked as being 'behind spindle poles' using a 3D Imaris Viewer. Lagging chromosomes were defined as a single kinetochore that was stuck and stretched between the separating mass of kinetochores during early anaphase. Chromosome bridges included cells with a kinetochore pair that was well separated but remained between the separating mass of kinetochores during early anaphase. Misalignments included cells that had at least one pair of kinetochores at the pole during anaphase, and the 'no error' phenotype was defined as a cell with absence of the aforementioned phenotypes. Multipolar cells (one out of 190) were not included in the analysis. Quantitative analysis of all parameters was performed using custom-made MATLAB (MatlabR2021a 9.10.0) scripts.

For live tracking of individual chromosomes, RPE1-hTERT dCas9-3×GFP were transduced with lentiviruses containing single-guide RNAs targeting chromosome 1 (ATGCTCACCT) and chromosome 9 (TGGAATGGAATGGAATGGAA). 24 h post transduction, cells were plated in an optical-quality, plastic, eight-well slide (IBIDI) at 50% confluency. After 16 h, asynchronous mitotic cells were treated with 62.5 nM Cpd-5 and immediately imaged using a ×1.4/40 NA oil PLAN Apochromat lens on a Zeiss Cell Observer microscope equipped with a AxioImager Z1 stand, a Hamamatsu ORCA-flash 4.0 camera and a Colibri 7 LED. Images were acquired every 2.5 min for 4.2 h. Videos were subsequently pro-cessed and analysed using ZEN software (Zeiss, v.3.3).

Chromosome 9 tracking and tethering experiments were performed on the spinning-disk system as previously described, with several adap-tations; 500 nM rapalog (Takara) was added 24 h before imaging of DLD1 cells and 62.5 nM Cpd-5 was added immediately before imaging. We used a ×1.20/60 NA water phase immersion oil lens, and 16 z-slices of 1 μm were imaged every 3 min overnight.

Cells were imaged at 37 °C in 5% $CO_2$ for all imaging experiments.

## MN-seq
RPE1-hTERT Flp-in cells were plated in a six-well plate at 40% confluency and treated with Cpd-5 or nocodazole for 16 h. Cancer cell lines were plated in a similar fashion, but were not treated with any drugs. Prepa-rations for FACS were performed similarly to the method described previously[33]. In short, cells were incubated on ice for 30 min under light with PBS/2% FBS and 12.5 μg ml$^{-1}$ EMA (ThermoFisher). EMA was washed four times using PBS, and (micro)nuclei were harvested from cells with the same nuclear staining buffer used for scKaryo-seq. EMA-negative and Hoechst-positive (micro)nuclei were sorted in bulk in a PCR strip containing mineral oil and stored at −20 °C for further processing. Library preparation was performed similarly to scKaryo-seq, but with several modifications. Every 5 μl of sorted (micro)nuclei was incubated with 5 μl of lysis buffer (final concentration, 0.02 U Proteinase K μl$^{-1}$ (NEB) in 1× CutSmart Buffer (NEB)) for 2 h at 55 °C and 10 min at 80 °C. Genomic DNA was digested by incubation of (micro)nuclei with 10 μl of digestion mix (final concentration, 0.5 U NLAIII μl$^{-1}$ (New England Biolabs) in 1× CutSmart Buffer) for 2 h at 37 °C, followed by 20 min at 65 °C. Genomic DNA fragments were subsequently ligated to adaptors by the addition of 20 μl of ligation mix (final concentration, 20 U μl$^{-1}$ T4 DNA ligase (New England Biolabs), 0.5 mM ATP (ThermoFisher) and 25 nM adaptor in 0.5× T4 DNA ligase buffer (New England Biolabs), with incubation at 16 °C overnight. After ligation, the remainder of library preparation, sequencing and analysis was performed as described for scKaryo-seq. To determine the percentage of reads per chromosome, all reads mapped to a specific chromosome were summed and normalized by dividing this by the number of bins for that specific chromosome. The percentage of reads for chromosome 10 in RPE1-hTERT cells was normalized using bulk-sequenced nuclei, because the q-arm of this chromosome is present in three copies.

## DamID

U2OS DamID sequencing data were generated in bulk from clonal cell lines stably expressing Dam-LaminB1 or untethered Dam protein. DamID data from Shield1-inducible DamID U2OS cells were derived by transfection of Dam-LaminB1 or Dam constructs (cloned into the pPTuner IRES2 vector (Clontech, Takara)), antibiotic resistance selection with 500 µg ml$^{-1}$ G418 (Gibco) and subsequent characterization of monoclonal cell populations. Selection of suitable clones was based on methylation concentrations at known LAD or iLAD genomic regions, measured by quantitative MboI-based PCR and DamID as previously described[54]. Stabilization of Dam proteins was achieved by the addition of Shield1 ligand (AOBIOUS) to the cell culture medium at 500 nM final concentration for 18–24 h before cell collection. Multiplexed DamID was performed as previously described[54] and sequenced on an Illumina NextSeq 500 platform (1× 50 bp). Raw reads were demultiplexed by their library-specific index and sample-specific DamID barcode, universal DamID adaptor sequence was trimmed with cutadapt (v.1.16) and reads were aligned to reference genome hg19 using bowtie2 (v.2.3.4). Reads mapping to annotated GATC sites were counted and aggregated in genomic bins of 100 kb. Computation of observed over expected values per bin was performed as previously described[55].

## Statistics

Statistical analyses were performed using GraphPad Prism software (v.8.4.3). Superplots were used in many of the graphs in which each colour represents a replicate, the small dots individual measurements and large dots the mean of each replicate.

## Reporting summary

Further information on research design is available in the Nature Research Reporting Summary linked to this paper.

## Data availability

Raw sequencing data can be found at the European Nucleotide Archive: PRJEB52892. Source data are available at https://doi.org/10.6084/m9.figshare.19779736. Previously published Dam- and Mad-Lamin B1 sequencing data for RPE1-hTERT can be found at GSM3904483, GSM3904484, GSM3904548 and GSM3904549, for hESC at GSM557443 and GSM557444, for TIG3-hTERT at GSM2030834, for HeLa at E-MTAB-6888, for K562 at GSM1612855 and GSM1612856 and for HT1080 at GSM984848.

## Code availability

Code used to generate figures from raw sequencing data can be found at https://github.com/sjklaasen/scKaryo-seq.git.

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

**Acknowledgements** We thank J. Vivié, L. Kester and D. Mooijman for help with scKaryo-seq, S. Tommouhi for help with organoid culturing and live imaging, the Hubrecht Flow Cytometry facility for help with sorting, the Hubrecht Imaging Centre for help with microscopy and the Utrecht Sequencing Facility for providing a sequencing service and data. We thank members of the Kops, Lens and Tolić laboratories for discussions and comments on the manuscript. Figure 3d and Extended Data Figs. 6f and 9 were created with Biorender.com. This study was funded by the Cancer Genomics Centre and the European Research Council (no. ERC-SyG 855158). The Kops, Lens and Kind laboratories are part of the Oncode Institute, which is partly funded by the Dutch Cancer Society (KWF Kankerbestrijding). The Tolić laboratory is funded by the Croatian Science Foundation (no. PZS-2019-02-7653) and European Regional Development Fund (nos. KK.01.1.1.04.0057 and KK.01.1.1.0004).

**Author contributions** S.J.K., R.H.v.J. and G.J.P.L.K. conceived the project. S.J.K. and G.J.P.L.K. wrote the manuscript. S.J.K. designed, performed and analysed all experiments unless specified otherwise. E.M.G. and R.H.v.J. set up the RPE1-hTERT cell synchronization procedure. R.H.v.J. performed and analysed RPE1-hTERT synchronization for filming and scKaryo-seq, and performed and analysed FISH on anaphase figures. J.F.M. generated the tetraploid RPE1 cell line. Live-tracking experiments for RPE1 cells in Fig. 3d–k were supervised by I.M.T. and performed and analysed by I.K. and V.Š. Live-tracking experiments for U2OS cells were performed by P.R. and S.K. and analysed by K.V. M.A.T. and S.G.d.V. designed, performed and analysed dCas9-imaging experiments. S.J.K. and M.A.T. performed and analysed dCas9-tethering experiments. K.d.L. performed and analysed U2OS DamID experiments. B.B. and F.F. provided help with setting up scKaryo-seq analysis. All authors provided input on the manuscript. J.K. and S.M.A.L. supervised experiments by K.d.L. and M.A.T./S.V., respectively, and G.J.P.L.K. supervised all aspects of the work.

**Competing interests** The authors declare no competing interests.

**Additional information**
**Correspondence and requests for materials** should be addressed to Geert J. P. L. Kops.

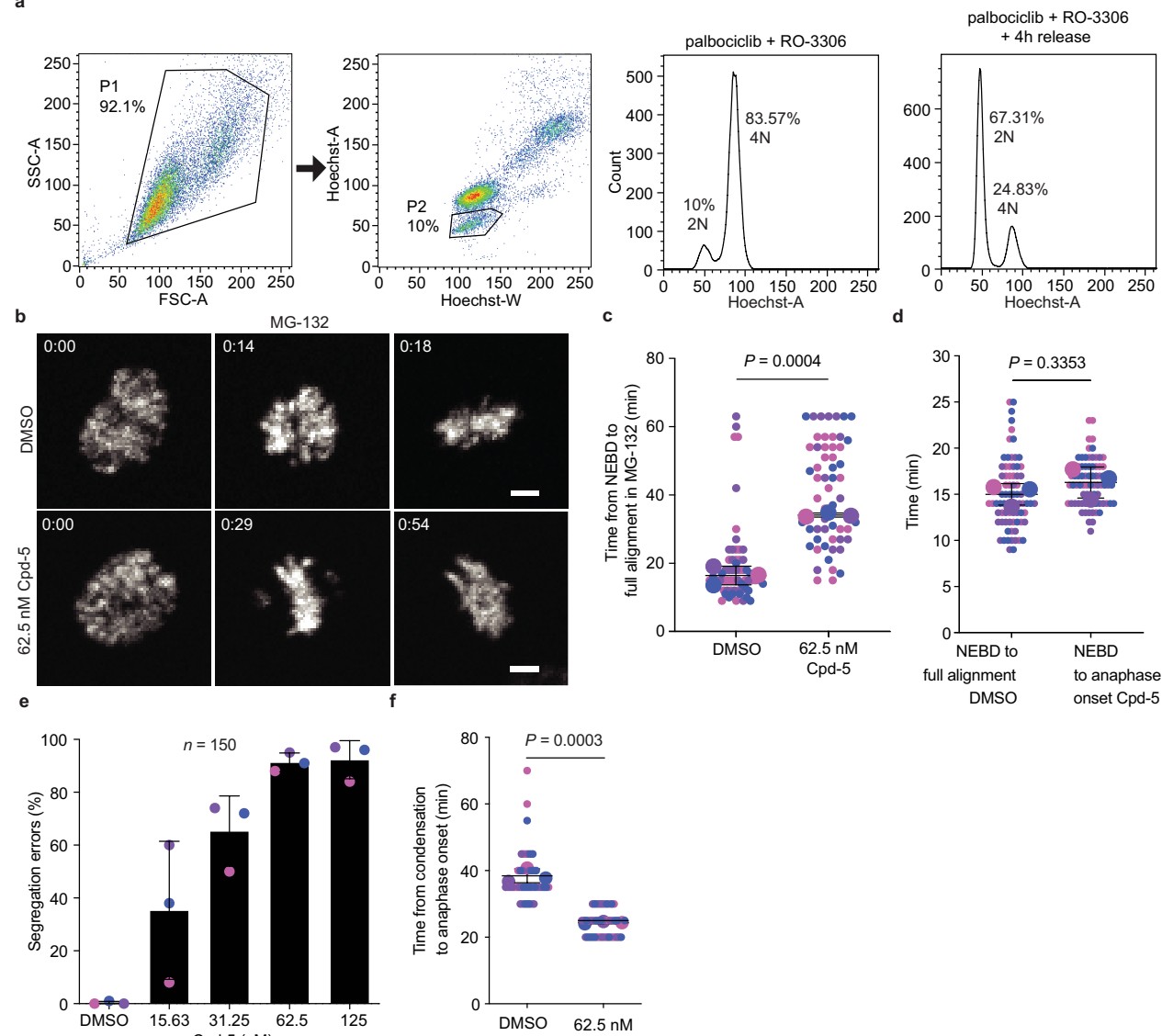

**Extended Data Fig. 1 | Characterization of RPE1-hTERT mis-segregations.**
**a**. Representative gating strategy for flow cytometry and the DNA content of
RPE1-hTERT cells synchronised using palbociclib and RO-3306. Fourth graph
on the right shows the DNA content 4 h after washing out RO-3306. **b,c**, Stills
(**b**) and quantification (**c**) of the time of alignment in synchronised and
MG-132-treated RPE1-hTERT H2B-mNeon cells (scale bar, 5 μm). Experiment
was performed in triplicate (mean ± s.d., two-tailed unpaired t-test, $n = 60$ and
58, respectively). **d**. Quantification of the time from condensation to full

alignment or anaphase onset in DMSO or Cpd-5, respectively. Experiment was
performed in triplicate (mean ± s.d., two-tailed unpaired t-test, $n = 75$).
**e**. Quantification of the segregation error percentages of RPE1-hTERT
H2B-mNeon treated with different concentrations of Cpd-5. Three
independent experiments were performed (mean ± s.e.m.). **f**. Quantification of
the time from condensation to anaphase onset of cells treated with DMSO or
62.5 nM Cpd-5. Three independent experiments were performed (mean ± s.d.,
two-tailed unpaired t-test, $n = 45$ cells).

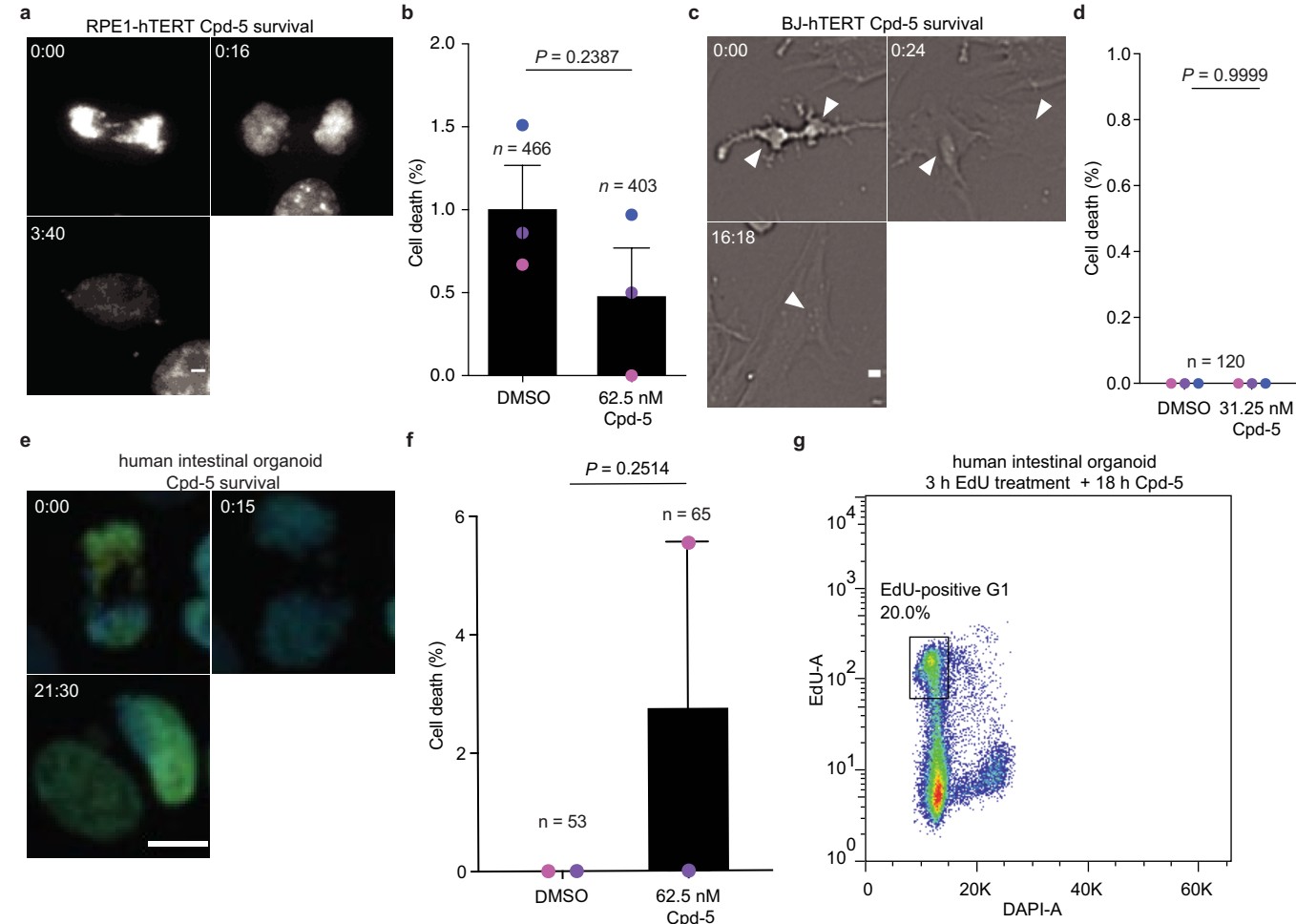

**Extended Data Fig. 2 | Characterization of survival after Cpd-5 treatment of multiple cell lines. a,b**, Representative stills of a RPE1-hTERT H2B-mNeon cell going through mitosis after synchronization and release in 62.5 nM Cpd-5 (**a**) and quantification of cell death (**b**). Three independent experiments were performed (scale bar, 5 μm, mean ± s.e.m., unpaired t-test). **c,d**, Stills of BJ-hTERT cells in 31.25 nM Cpd-5 (**c**) and quantification of cell death (**d**). Experiment was performed in triplicate (scale bar, 5 μm, mean ± s.e.m., two-tailed Fisher's exact test, $n$ = 120 daughter cells per condition). **e,f**, Stills (**e**) and quantification (**f**) of human intestinal organoids going through mitosis in the presence of Cpd-5 (scale bar, 5 μm). Two independent experiments were performed (mean ± s.e.m., two-tailed Fisher's exact test). **g**. Graph illustrating sorting strategy of human intestinal organoids to enrich for cells with aneuploidies. Organoids were treated with EdU for 3h, washed and treated overnight with 62.5 nM Cpd-5. EdU-positive G1 cells were sorted.

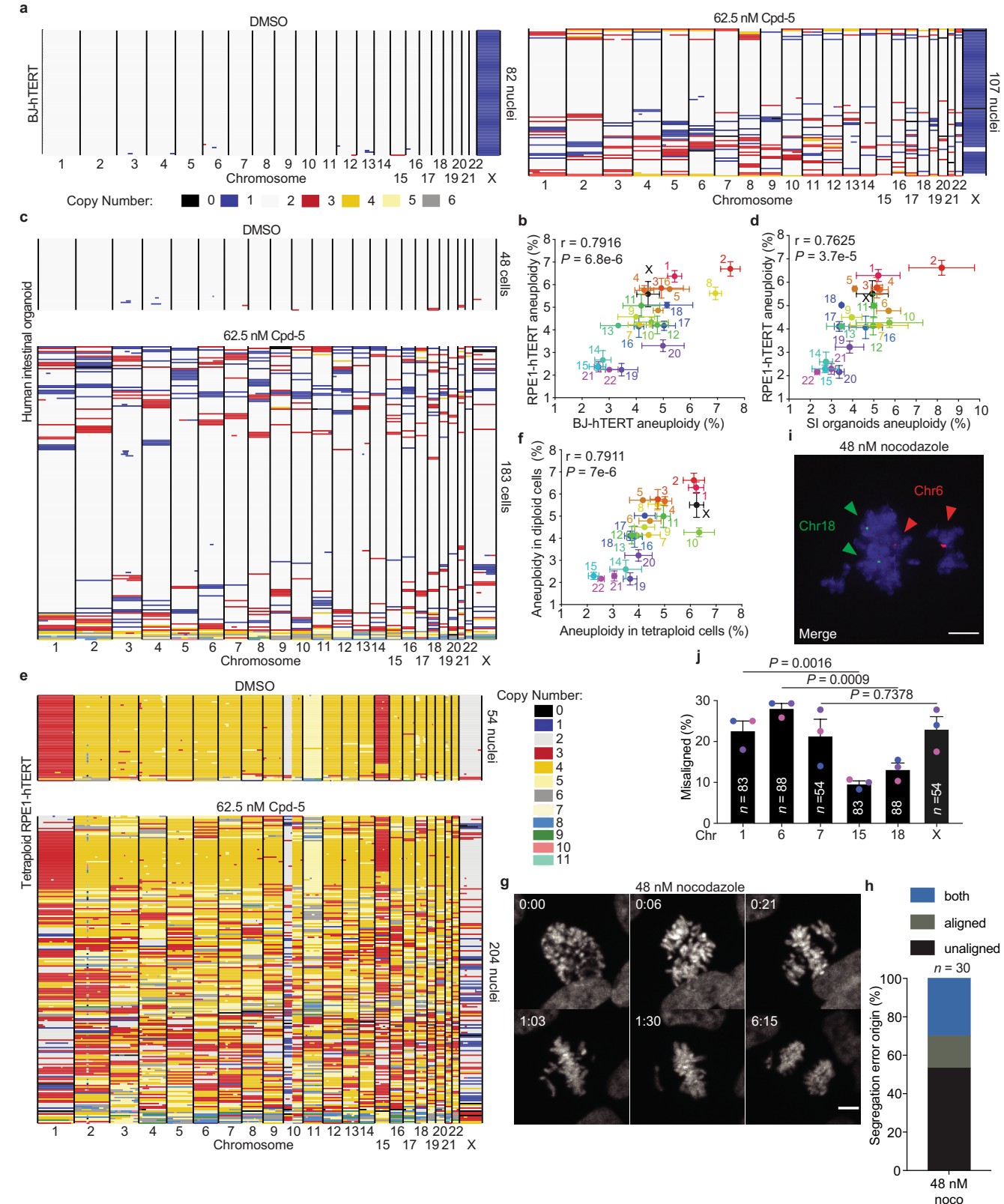

**Extended Data Fig. 3** | See next page for caption.

**Extended Data Fig. 3 | Similar mis-segregation frequencies in different cell lines and after different perturbations. a**. Representative scKaryo-seq replicate of BJ-hTERT cells treated for 16 h with Cpd-5. **b**. Plot comparing aneuploidy percentages of RPE1-hTERT cells after Cpd-5 treatment (see Fig. 1c) and quantification of BJ-hTERT aneuploidy percentages (mean ± s.e.m., two-tailed Pearson correlation coefficient, $n$ = 180 aneuploid cells). Three independent experiments were performed. **c**. Representative scKaryo-seq replicate of a human intestinal organoid line treated with or without Cpd-5 for 16 h. **d**. Plot comparing aneuploidy percentages of RPE1-hTERT cells (see Fig. 1c) and human intestinal organoid aneuploidy percentages after Cpd-5 treatment. (mean ± s.e.m., two-tailed Pearson correlation coefficient, $n$ = 217 aneuploid cells). Two independent organoid lines originating from the duodenum and ileum were used for this experiment. Chromosome 8 was not included in the analysis, because this chromosome was heterogeneously aneuploid in the ileum organoid line. **e**. Representative scKaryo-seq replicate of tetraploid RPE1-hTERT cells treated with and without Cpd-5 for 24 h. **f**. Plot comparing aneuploidy percentages of Fig. 1c versus quantification of the aneuploidy percentages after Cpd-5 treatment in tetraploid RPE1-hTERT cells (mean ± s.e.m., two-tailed Pearson correlation coefficient, n = 180 aneuploid cells). Experiment was performed in triplicate. **g,h**, Stills (**g**) and quantification (**h**) looking at the location of mis-segregating chromosomes just before anaphase onset after 48 nM nocodazole treatment (scale bar, 5 μm). Experiment was performed in triplicate ($n$ = 30 segregation error events). **i,j**, Representative FISH images (**i**) and quantification (**j**) of RPE1-hTERT cells synchronised as before and released in mitosis with 48 nM nocodazole (scale bar, 5 μm). Measured is the percentage of misaligned FISH probes (mean ± s.e.m., two-tailed Fisher's exact test). Three independent experiments were performed.

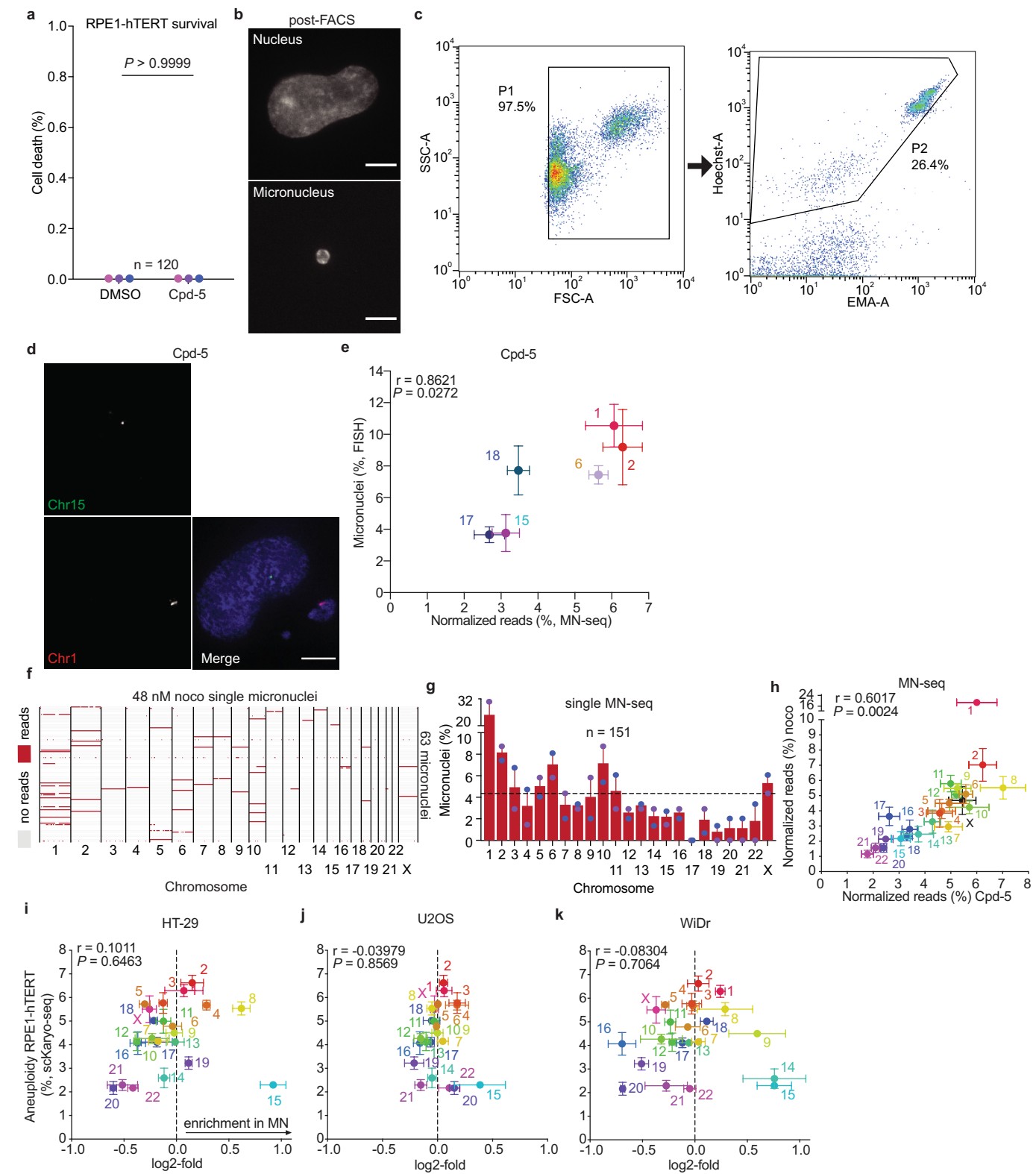

**Extended Data Fig. 4 | MN-seq characterization and micronucleus content bias in nocodazole-treated and cancer cells. a.** Live-cell imaging of RPE1-hTERT cells to determine survival when treated for 16 h with 62.5 nM Cpd-5. Experiment was performed in triplicate (mean ± s.e.m., two-tailed Fisher's exact test). **b.** Representative images of a FAC-sorted nucleus and micronucleus (scale bar, 5 μm). **c.** Gating strategy for micronuclei. **d,e,** Representative FISH images (**d**) and quantification (**e**) of micronucleated RPE1-hTERT cells treated with Cpd-5 for 16 h (scale bar, 5 μm). The graph shows the percentage of micronuclei containing a certain chromosome (s.e.m., two-tailed Pearson correlation coefficient, 1/15 (*n* = 371 micronuclei), 2/17 (*n* = 277 micronuclei)

and 6/18 (*n* = 376 micronuclei)). Three independent experiments were performed. **f,g,h,** as in Fig. 2b–d but instead cells were treated with low nocodazole (single micronuclei; mean ± s.e.m, two independent experiments, *n* = 151, bulk micronuclei; mean ± s.e.m., four independent experiments, two-tailed Pearson correlation coefficient, *n* = ~8000 micronuclei). **i,j,k,** Log2-fold enrichment of chromosomes in MN determined from bulk MN-seq data sorted from three chromosomally unstable cancer cell lines (mean ± s.e.m., two-tailed Pearson correlation coefficient, *n* = ~1500 MN for HT-29 and WiDr and 6000 for U2OS). Three independent experiments were performed for each cell line.

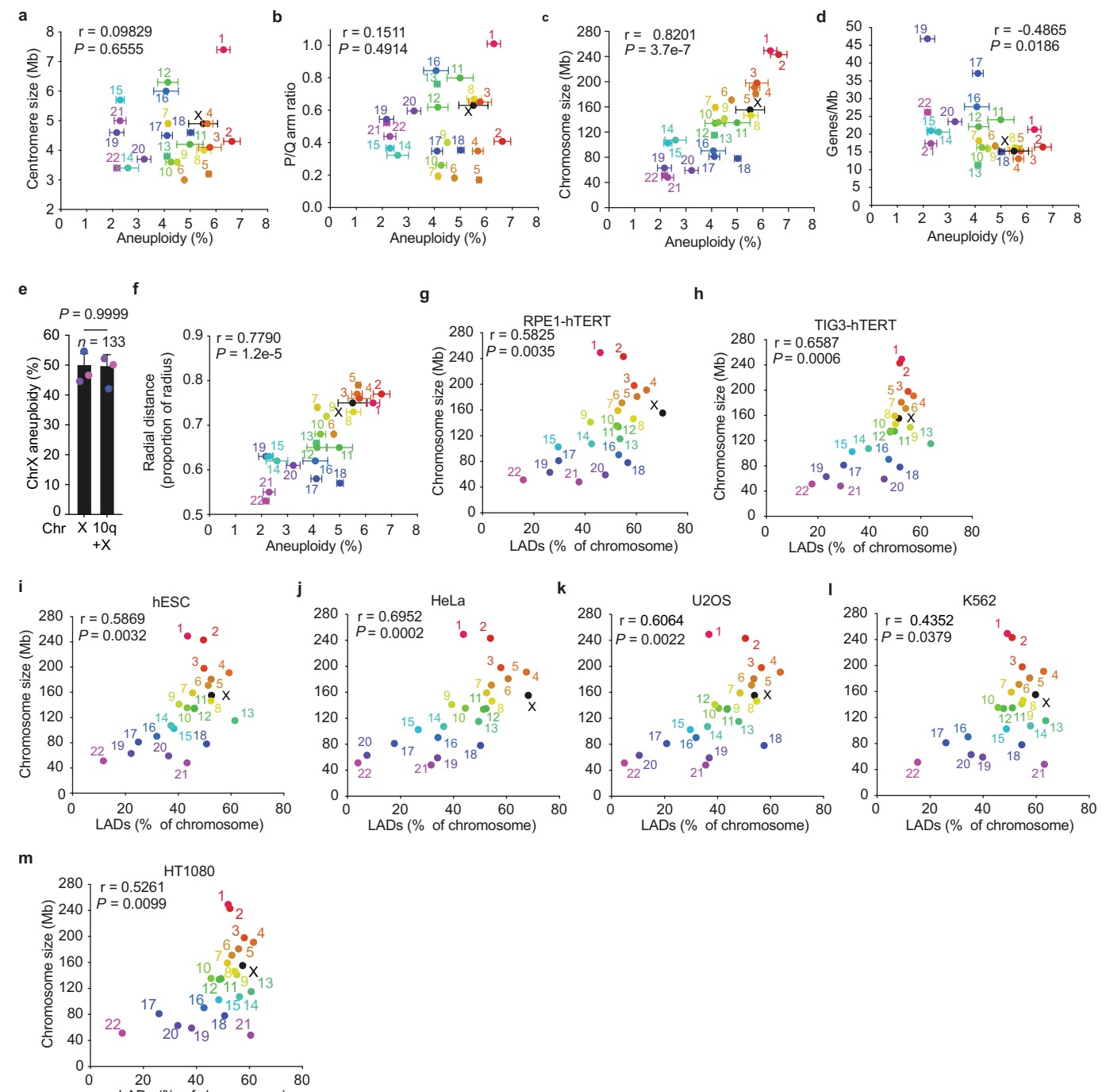

**Extended Data Fig. 5 | Mis-segregation frequency correlates with chromosome organization and organization is similar between cell lines. a,b,c,d**, Plots comparing aneuploidy percentages after Cpd-5 treatment (see Fig. 1c) in RPE1-hTERT cells to centromere sizes (**a**), ratio of P- over Q arm sizes (**b**), chromosome sizes (**c**), and gene densities (**d**) (two-tailed Pearson correlation coefficient). **e**. Quantification of aneuploidy percentages of both X chromosomes in RPE1-hTERT cells after Cpd-5 treatment (mean ± s.e.m.,

two-tailed Fisher's exact test, $n$ = 133 cells with an X chromosome aneuploidy). **f**. Plot comparing aneuploidy percentages after Cpd-5 treatment versus previously determined radial distances (two-tailed Pearson correlation coefficient)[37]. **g,h,i,j,k,l,m**, Plots comparing the percentage of LADs per chromosome for indicated cell lines to chromosome size (two-tailed Pearson correlation coefficient).

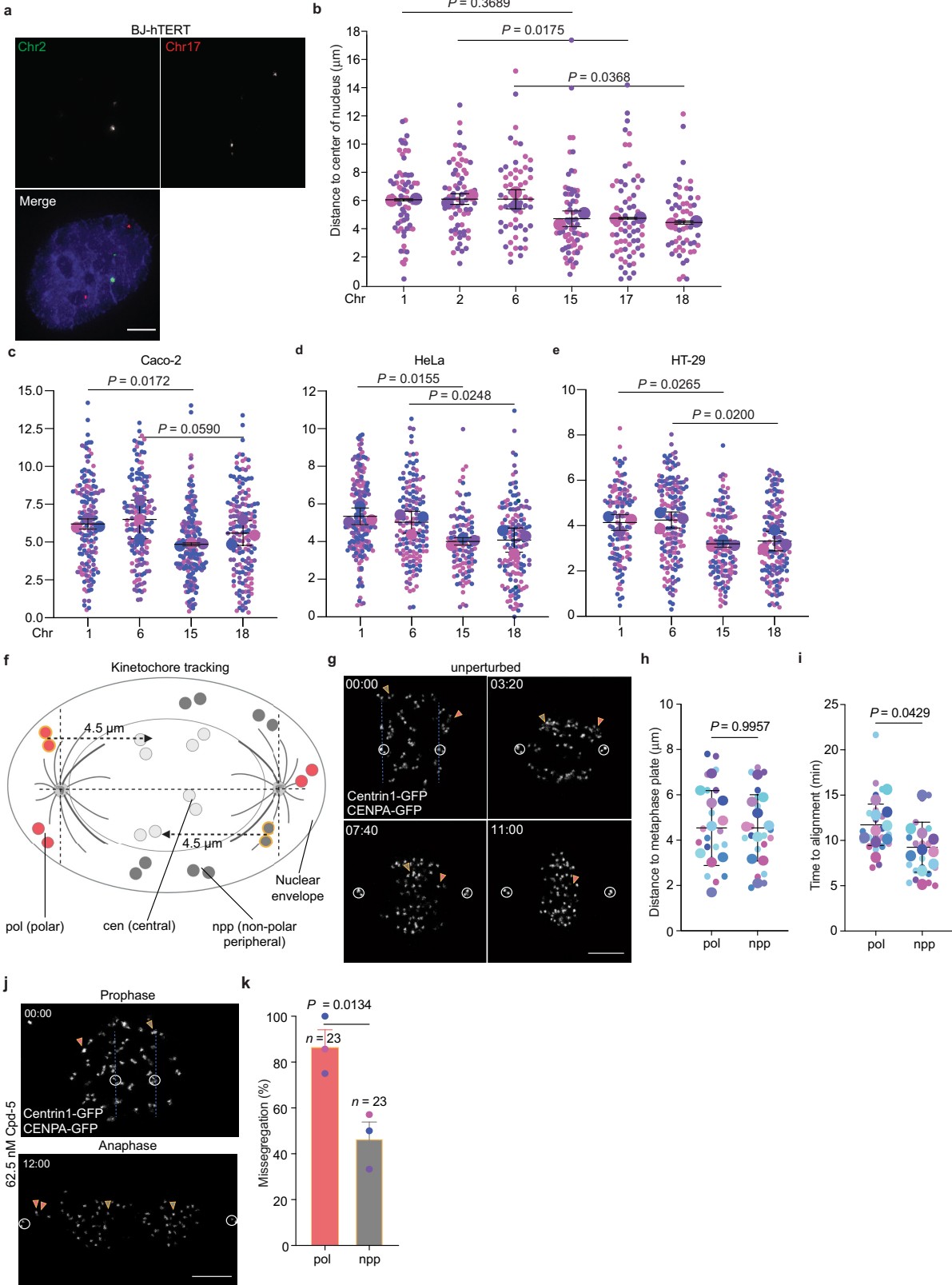

**Extended Data Fig. 6** | See next page for caption.

**Extended Data Fig. 6 | Chromosomes behind the poles are more likely to mis-segregate. a,b**, Representative images (**a**) and quantification (**b**) of the distance from centromeric FISH probes to the centre of the nucleus in BJ-hTERT cells (scale bar, 5 μm, mean ± s.d., $n > 70, 56, 68, 70$ and 68 chromosomes, respectively). Data is pooled from two independent experiments. **c,d,e**, Distance of centromeric FISH probes to the centre of the nucleus in three cancer cell lines (mean ± s.d., two-tailed ratio t-test, $n = 133, 112, 81$ and 108; $n = 125, 161, 134$ and 153; $n = 160, 151, 185, 160$ chromosomes, respectively). Three independent experiments were performed. **f**. Schematic depicting the strategy to follow kinetochores with a similar distance to the metaphase plate (yellow circles). **g,h,i**, Representative stills (**g**) and quantification (**h-i**) of time to alignment for matched kinetochores based on them having the same distance to the metaphase plate in RPE1-hTERT CENPA-GFP Centrin1-GFP cells (scale bar, 5 μm). White circles mark the centrosomes. Experiment was performed 10 times (mean ± s.d., unpaired t-test, $n = 21$). **j,k**, As in **e-g**, but instead cells were treated with 62.5 nM Cpd-5 and mis-segregations were measured (scale bar, 5 μm). Experiment was performed in triplicate (mean ± s.e.m., Fisher's exact test).

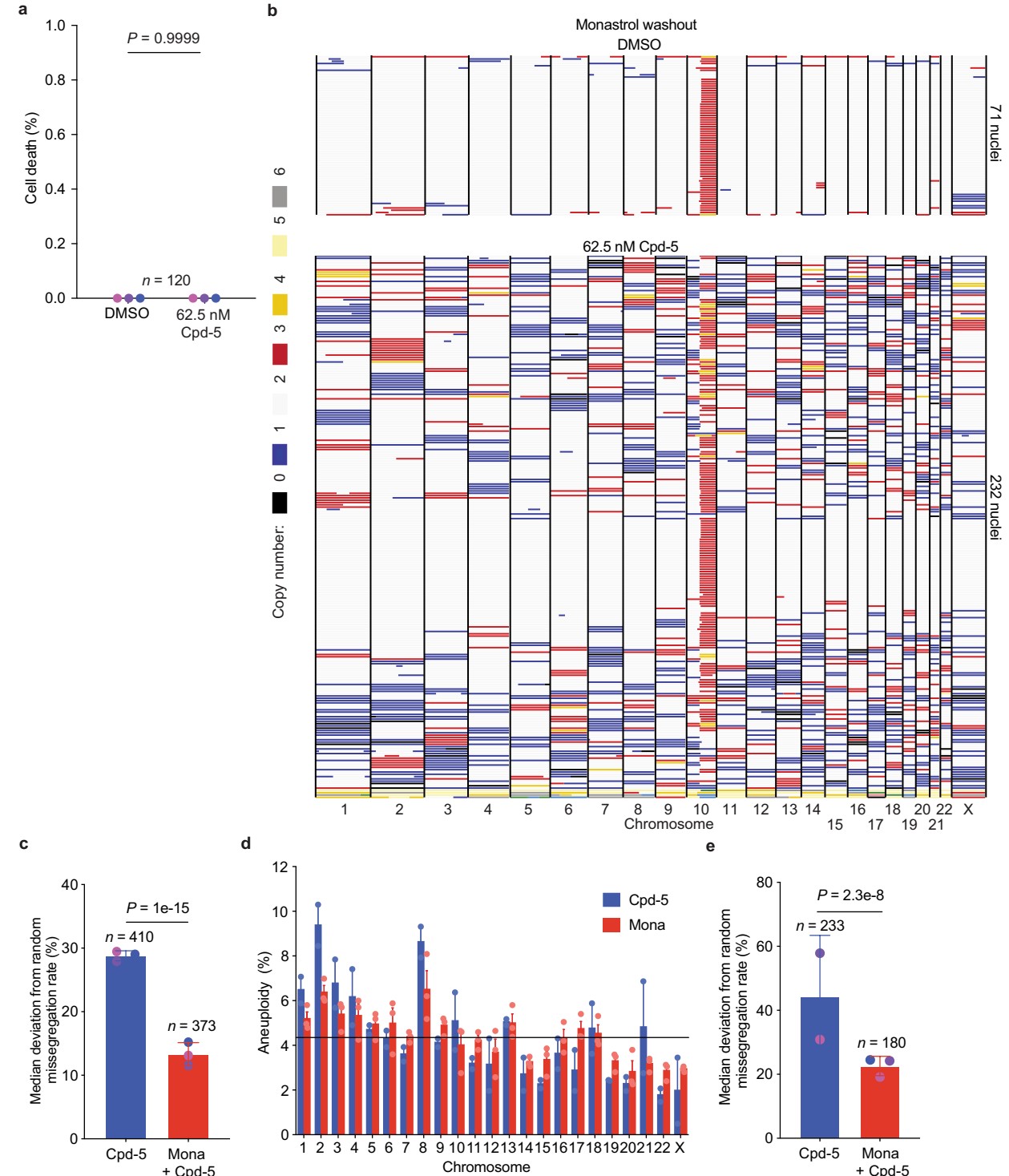

**Extended Data Fig. 7 | Randomizing chromosome organization with monastrol decreases mis-segregation bias a.** Quantification of cell survival of RPE1-hTERT cells synchronised, treated for 4 h with monastrol, followed by a washout and shake-off. Experiment was performed three times (scale bar, 5 μm, mean ± s.e.m., two-tailed Fisher's exact test, *n* = 120 daughter cells per condition). **b.** Representative scKaryo-seq results of RPE1-hTERT cells undergone a monastrol washout. **c.** Quantification of the change in the observed mis-segregation rate compared to expected (4.3%) for Cpd-5 only treated RPE1-hTERT cells versus monastrol plus Cpd-5 treated ones (mean ± s.e.m., two-tailed Fisher's exact test). **d.** Quantification of aneuploidy levels of HCT116 either treated with Cpd-5 only or a combination of monastrol and Cpd-5 as described before. Experiment was performed twice for the Cpd-5 treated cells and three times for the monastrol treated ones (mean ± s.e.m., *n* = 233 and 180 aneuploid cells, respectively). **e.** As in c, but for HCT116 cells (mean ± s.e.m., two-tailed Fisher's exact test).

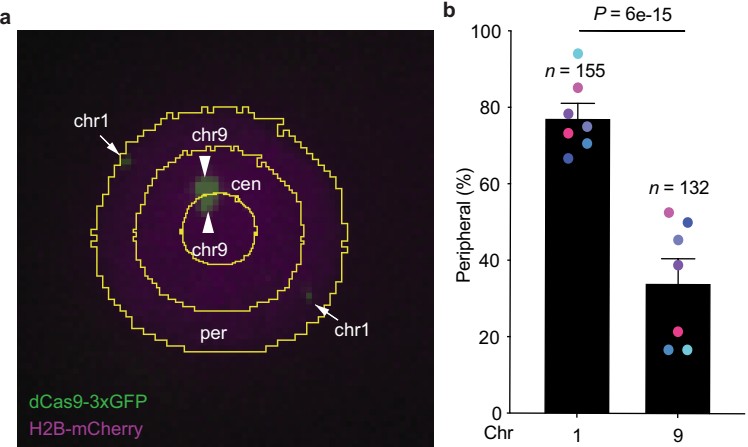

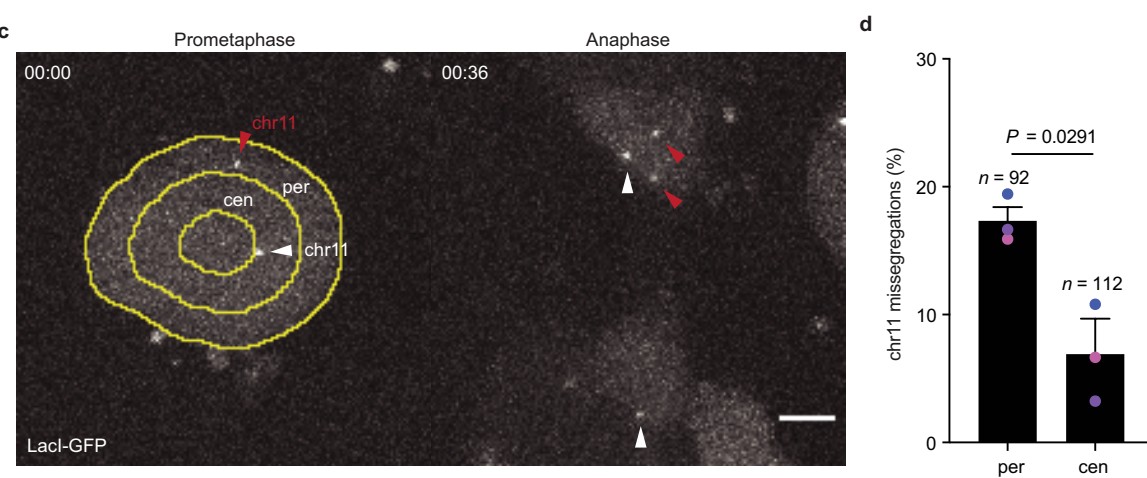

**Extended Data Fig. 8 | Chromosomes in the periphery of the nucleus mis-segregate more frequently independent of identity. a,b,** Representative still (**a**) and quantification (**b**) of the percentage of Cas9-tagged peripheral chromosomes just before mitosis onset (mean ± s.e.m., two-tailed Fisher's exact test). Only chromosomes in the most outer ring were considered peripheral. **b,c,** Representative stills (**c**) and quantification (**d**) of chr11 mis-segregation rate of LacI-GFP-expressing HT1080 cells containing a LacO array in chr11 (scale bar, 5 μm). Red arrowheads follow a mis-segregating chromosome pair, while white ones follows a properly segregating pair. Experiment was performed in triplicate (mean ± s.e.m., two-tailed Fisher's exact test).

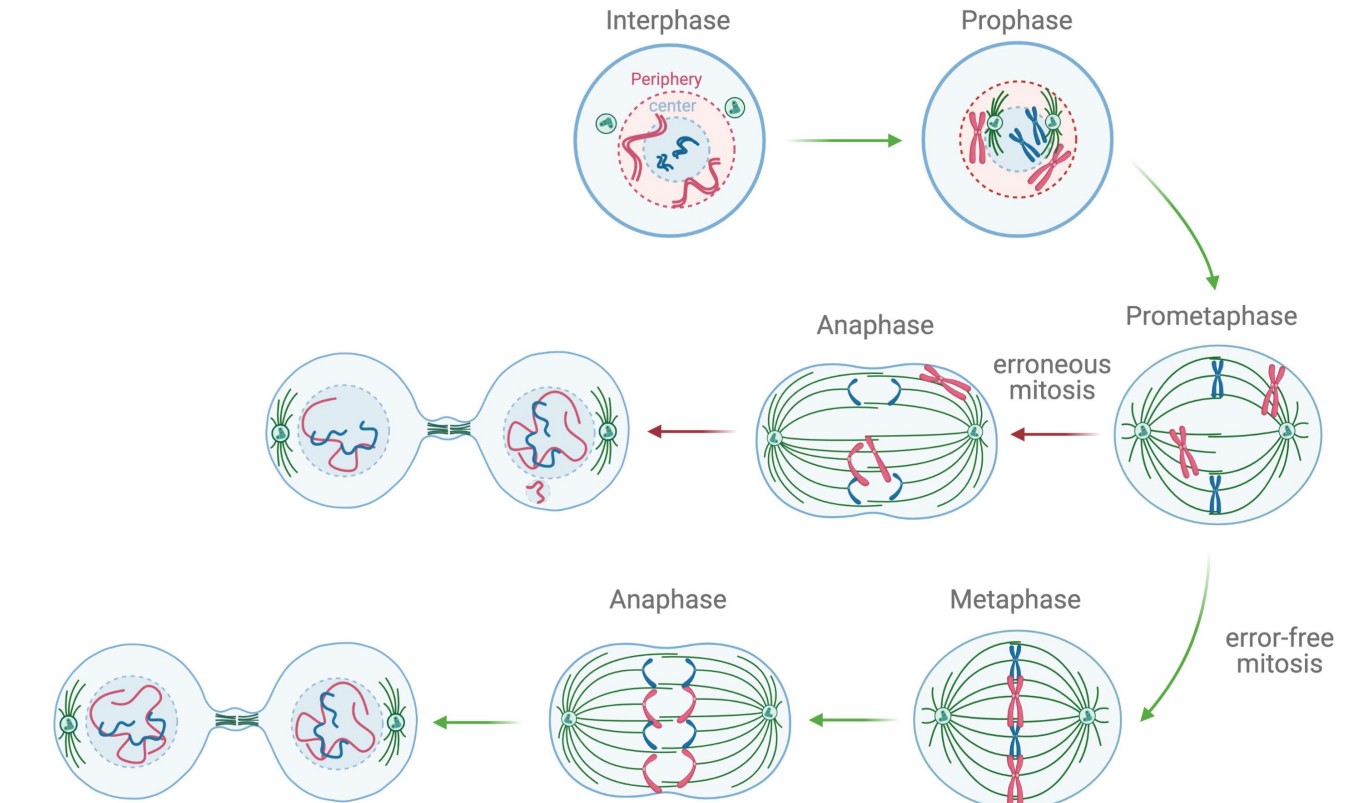

**Extended Data Fig. 9 | Graphical representation of a cell going through an error-free mitosis (green arrows) or an erroneous one (red arrows).** Red chromosomes start out peripheral, while blue ones are more central.

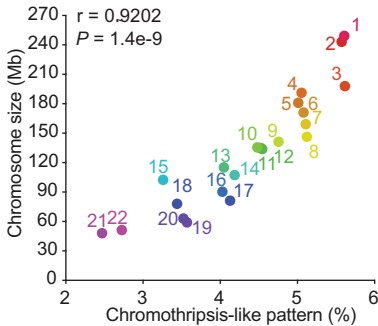

**Extended Data Fig. 10 | Chromothripsis-like patterns correlate with chromosome size.** Plot comparing chromosome size and chromothripsis-like patterns percentages per chromosome calculated from previously published data (two-tailed Pearson correlation coefficient).

# Reporting Summary

## Statistics

For all statistical analyses, confirm that the following items are present in the figure legend, table legend, main text, or Methods section.

| n/a | Confirmed | |
|---|---|---|
| ☐ | ☒ | The exact sample size (*n*) for each experimental group/condition, given as a discrete number and unit of measurement |
| ☐ | ☒ | A statement on whether measurements were taken from distinct samples or whether the same sample was measured repeatedly |
| ☐ | ☒ | The statistical test(s) used AND whether they are one- or two-sided *Only common tests should be described solely by name; describe more complex techniques in the Methods section.* |
| ☒ | ☐ | A description of all covariates tested |
| ☐ | ☒ | A description of any assumptions or corrections, such as tests of normality and adjustment for multiple comparisons |
| ☐ | ☒ | A full description of the statistical parameters including central tendency (e.g. means) or other basic estimates (e.g. regression coefficient) AND variation (e.g. standard deviation) or associated estimates of uncertainty (e.g. confidence intervals) |
| ☐ | ☒ | For null hypothesis testing, the test statistic (e.g. *F*, *t*, *r*) with confidence intervals, effect sizes, degrees of freedom and *P* value noted *Give P values as exact values whenever suitable.* |
| ☒ | ☐ | For Bayesian analysis, information on the choice of priors and Markov chain Monte Carlo settings |
| ☒ | ☐ | For hierarchical and complex designs, identification of the appropriate level for tests and full reporting of outcomes |
| ☐ | ☒ | Estimates of effect sizes (e.g. Cohen's *d*, Pearson's *r*), indicating how they were calculated |

*Our web collection on statistics for biologists contains articles on many of the points above.*

## Software and code

Policy information about availability of computer code

| | |
|---|---|
| Data collection | Imaging data were collected using SoftWorx (Applied Precision/GE Healthcare, version 6.5.2), NIS-Elements (Nikon, version 5.30.04), Prairie View (version 5.4.64.500); Imspector (Abberior Instruments GmbH, version 16.3) or ZEN (Zeiss, version 3.3) <br> Flow cytometric data were collected using FACSDiva (BD Biosciences, version 9.0.1) <br> Single-end sequencing was performed on a Nextseq500 (50bp or 75bp) or NextSeq2000 (100bp) (Illumina, San Diego, CA, USA) |
| Data analysis | Raw reads were mapped to hg38 using bwa aln (version 0.7.12). Only reads starting with an NLAIII cutter site and with unique chromosome coordinates plus unique UMI were kept using python (version 2.7.5). <br> cutadapt (version 1.16) <br> bowtie2 (version 2.3.4) <br> Code to generate figures from raw sequencing data can be found at https://github.com/sjklaasen/scKaryo-seq.git. <br><br> Sequencing data were visualized and copy numbers were determined using Aneufinder (version 1.2.0) in RStudio (version 1.4.1717). <br><br> Graphpad Prism 8 was used for statistical analyses and data visualization (version 8.4.3) <br><br> Fiji ImageJ was used for imaging analyses (version 2.0.0) <br><br> FACS data was analyzed on Flowjo (version 10.6.1) <br><br> For kinetochore tracking experiments: <br> Data analysis and tracking: Fiji ImageJ (version 1.53f51/1.53s30/1.53r; Low light) <br> tracking tool plugin (version 0.10); Bio Format plugin (version 6.7.0); ImarisViewer (version 9.8.0) |

Quantification: MatlabR2021a (version 9.10.0)

For manuscripts utilizing custom algorithms or software that are central to the research but not yet described in published literature, software must be made available to editors and reviewers. We strongly encourage code deposition in a community repository (e.g. GitHub). See the Nature Portfolio guidelines for submitting code & software for further information.

## Data

Policy information about availability of data

All manuscripts must include a data availability statement. This statement should provide the following information, where applicable:
- Accession codes, unique identifiers, or web links for publicly available datasets
- A description of any restrictions on data availability
- For clinical datasets or third party data, please ensure that the statement adheres to our policy

Raw sequencing data can be found at the European Nucleotide Archive: PRJEB52892.

Source data are available at https://doi.org/10.6084/m9.figshare.19779736.

Previously published Dam- and Mad-Lamin B1 sequencing data for RPE1-hTERT can be found at GSM3904483, GSM3904484, GSM3904548 and GSM3904549, for hESC at GSM557443 and GSM557444, for TIG3-hTERT at GSM2030834, for HeLa at E-MTAB-6888, for K562 at GSM1612855, GSM1612856 and for HT1080 at GSM984848.

# Field-specific reporting

Please select the one below that is the best fit for your research. If you are not sure, read the appropriate sections before making your selection.

☒ Life sciences    ☐ Behavioural & social sciences    ☐ Ecological, evolutionary & environmental sciences

For a reference copy of the document with all sections, see nature.com/documents/nr-reporting-summary-flat.pdf

# Life sciences study design

All studies must disclose on these points even when the disclosure is negative.

| | |
|---|---|
| Sample size | No statistical methods were used to predetermine sample size. Between 20-200 cells were analyzed per replicate, depending on type of experiment (live cell tracking, FISH, sequencing). The size of the sample was chosen to offer sufficient statistical power. |
| Data exclusions | scKaryo-seq data was excluded manually if copy number states could not be easily discerned or did not fit ploidy state after FACS. Micronuclei were excluded from single MN-seq analysis if they contained five or more chromosomes. |
| Replication | All attempts of replication were successful. Experiments were repeated at least twice. |
| Randomization | No randomization was performed. To counter batch effects, both conditions were usually sorted into the same plate. |
| Blinding | Data collection during FISH imaging data collection was blinded by only looking at the DAPI signal when selecting cells. For imaging and sequencing experiments, blinding was not possible because conditions could be easily identified from the sequencing or imaging data itself. |

# Reporting for specific materials, systems and methods

We require information from authors about some types of materials, experimental systems and methods used in many studies. Here, indicate whether each material, system or method listed is relevant to your study. If you are not sure if a list item applies to your research, read the appropriate section before selecting a response.

## Materials & experimental systems

| n/a | Involved in the study |
|---|---|
| ☒ | ☐ Antibodies |
| ☐ | ☒ Eukaryotic cell lines |
| ☒ | ☐ Palaeontology and archaeology |
| ☒ | ☐ Animals and other organisms |
| ☒ | ☐ Human research participants |
| ☒ | ☐ Clinical data |
| ☒ | ☐ Dual use research of concern |

## Methods

| n/a | Involved in the study |
|---|---|
| ☒ | ☐ ChIP-seq |
| ☐ | ☒ Flow cytometry |
| ☒ | ☐ MRI-based neuroimaging |

# Eukaryotic cell lines

Policy information about cell lines

| | |
|---|---|
| Cell line source(s) | RPE1-hTERT cells were a gift from the Prasad Jallepalli lab.<br>HeLa cells were a gift from the Michiel Vermeulen lab.<br>BJ-hTERT cells were a gift from the Rene Medema lab.<br>U2OS cells were a gift from the Susanne Lens lab.<br>DLD1 cells were a gift from the Daniella Cimini lab.<br>Human intestinal organoids, HCT116, Caco-2, HT-29 and Widr cell lines were a gift from the Hans Clevers lab.<br>RPE1-hTERT CENPA-GFP and Centrin1-GFP cells were a gift from the Alexey Khodjakov lab.<br>U2OS CENPA-GFP mCherry-α-tubulin photoactivatable-GFP-α-tubulin were a gift from the Helder Maiato lab.<br>HT1080 cells containing a LacO-array in chromosome 11 were a gift from the Wendy Bickmore lab. |
| Authentication | RPE1, BJ and HCT116 cell lines were verified based on the karyotypes as determined by scKaryo-seq. |
| Mycoplasma contamination | All lines were tested negative for mycoplasma. |
| Commonly misidentified lines<br>(See ICLAC register) | No commonly misidentified lines were used. |

# Flow Cytometry

## Plots

Confirm that:

☒ The axis labels state the marker and fluorochrome used (e.g. CD4-FITC).

☒ The axis scales are clearly visible. Include numbers along axes only for bottom left plot of group (a 'group' is an analysis of identical markers).

☒ All plots are contour plots with outliers or pseudocolor plots.

☒ A numerical value for number of cells or percentage (with statistics) is provided.

## Methodology

| | |
|---|---|
| Sample preparation | Single G1 nuclei of RPE1-hTERT Flp-in cells or single nuclei of BJ-hTERT cells were isolated as described before (Bolhaqueiro et al., 2019 (Nat. Gen.). In short, cells were treated for around 15 min on ice with a nuclear staining buffer containing 100 mM Tris-HCl pH 7.5, 154 mM NaCl, 1 mM CaCl2, 0.5 mM MgCl2, 0.2% BSA, 0.1% NP40 (v/v), 1 ug/mL Hoechst 34580.<br><br>Small intestinal cells treated with EdU were fixed using 70% ice-cold ethanol. Ethanol was removed by one wash with PBS and cells were incubated for 10 min with the Click-iT reaction cocktail (see Click-iT EdU proliferation assay). The reaction cocktail was washed away and replaced with a PBS/DAPI mix. Single G1 nuclei in case of ZM447439 or EdU-positive G1 cells were sorted in 384-well plates.<br><br>For MN-seq, cells were incubated on ice for 30 min under light with PBS/2% FBS and 12.5 μg/ml EMA (Thermofisher). EMA was washed 4x using PBS and (micro)nuclei were harvested from the cells using the same nuclear staining buffer used for scKaryo-seq. EMA-negative and Hoechst-positive (micro)nuclei were sorted in bulk in a PCR strip containing mineral oil and stored at -20 °C for further processing. |
| Instrument | BD FACSJazz (cat. num. 655489), FACSAria II SORP (serial number P58000001) and FACSAria FUSION SORP (serial number R658282P4001) (BD, Franklin Lakes, NJ, USA) |
| Software | Flow cytometric data were collected using FACSDiva (BD Biosciences, version 9.0.1)<br>Flowjo (version 10.6.1) |
| Cell population abundance | Single cell or nucleus sorting purity was not explicitly determined, because a higher purity would only increase the percentage of aneuploid cells, but not influence results. Micronucleus sorting purity was visually confirmed after sorting using microscopy. |
| Gating strategy | FSC/SSC and Hoechst were used to gate for cells and singlets (Extended Data Fig. 1a). EdU-positive G1 cells were selected based on EdU and DAPI-staining (Extended Data Fig. 2g). Micronuclei with a Hoechst intensity 10x below G1 were gated (Fig. 2a and Extended Data Fig. 4c). |

☒ Tick this box to confirm that a figure exemplifying the gating strategy is provided in the Supplementary Information.

