## [Peer Review File · Nature]

Manuscript Title: Nuclear chromosome locations dictate segregation error frequencies

Reviewer Comments & Author Rebuttals

Reviewer Reports on the Initial Version:

Referees' comments:

Referee #1 (Remarks to the Author):

In the submitted manuscript, Klaasen et al. demonstrate that human cells undergoing segregation errors show a bias for certain chromosomes. This is true in multiple cell lines (RPE1-hTERT, BJ-hTERT, intestinal organoids); and true irrespective of the particular method used to induce errors (MPS1 inhibition, Aurora B inhibition, nocodazole, "naturally occurring" micronuclei in Caco-2 and HeLa cells). These biases can be observed by multiple methods: 1) by assessing aneuploidies by single-cell sequencing (scKaryo-seq), 2) by chromosome-specific FISH, 3) or by sequencing of isolated micronuclei (MN-seq). While trying to determine the source of this bias, the authors observe that error probabilities correlate with the position of the chromosome in the interphase nucleus. This is assessed by correlating error rates with existing data on chromosome position and lamin-associated domain (LAD) frequency. The authors corroborate this finding with two live-cell imaging-based approaches to label and follow peripheral versus central chromosomes: 1) photoconversion of chromosomes labeled with H2B-Dendra2, and 2) live-cell imaging of specific chromosomes with dCas9. The authors go on to show that randomizing chromosome positions in mitosis (with monastrol) leads to a reduction of the chromosome segregation bias. These findings are important to the field because several different mechanisms for the previously identified chromosome-specific segregation bias have been proposed.

This paper reports quality data on an interesting topic and has both strengths and weaknesses. The conclusion that missegregation rates vary across chromosomes—although not novel- is well supported by the data and is strengthened by the use of complementary methods and model systems. The general quality of the work is high. On the negative side, this paper suggests a new mechanism for missegregation bias, but it is not exclusive with previously proposed mechanisms. Indeed, as described below, some experiments in this paper suggest that chromosome size may be an even bigger factor than chromosome position. Therefore, the degree to which this is a major advance over prior work can be debated. Moreover, the conclusion that chromosome position explains missegregation bias needs better experimental support. Finally, although there is some evidence that biased missegregation is evident in cancer genomes, the effects are fairly small, and selection seems likely to be the main constraint.

Major points

The main novel finding of this paper is that missegregation rates are dictated by a chromosome's nuclear position. However, most of the data supporting this model are correlative, not causative. The one experiment where chromosome position is experimentally perturbed is the one in which monastrol is used to randomize chromosome positions during mitosis. However, these experiments raise the following points:

(1) In previous papers (Worrall et al, Cell Reports 2018; Drpic et al, Current Biology 2018), monastrol treatment, led to an increase in biased segregation, not the decrease reported here. In the case of the Worrall et al paper, they used of the same tissue culture cells studied in the current manuscript. Perhaps this could be explained by the use of monastrol followed by MPS1 inhibition (this study), as opposed to monastrol alone (Worrall). However, this is an important point that

needs to be clarified experimentally. The authors should determine if monastrol washout alone still causes biased missegregation, despite randomizing of chromosome position. If this were to be the case, absent another explanation, this finding would contradict the authors' model.

(2) Although there is a correlation between interphase chromosome position and missegregation frequency, it is not perfect. This reviewer wonders if the correlation could be strengthened if the author looked directly at chromosome position in mitosis and correlated it with missegregation rates (e.g., using H2B-Dendra2).

(3) The strongest direct experimental evidence for position causing missegregation comes from the monastrol washout experiment. But it remains possible that this global disruption of spindle function might introduce its own set of segregation biases. It would improve the manuscript if it were possible to alter chromosome position without globally disrupting spindle assembly. Perhaps the authors could adapt engineered systems where individual chromosomes can be tethered to the nuclear periphery (Finlan et al PLoS Genetics 2008).

(4) In addition to the monastrol experiment, the authors employ several methods to assess the correlation between interphase chromosome position and missegregation rates. However, chromosome position does not, in fact, fully explain error rates: this point should be better emphasized in the text. For example, aneuploidy rates in RPE-1 cells actually correlate better with chromosome size (Fig. S7C, $r = 0.8201$) than with LAD frequency (Fig. 3A, $r = 0.7611$) or chromosome position (Fig. S7F, $r = 0.7790$). Additionally, in Fig. 3H, the distributions of the "distance to center" measurement for the "Error" vs. "No error" populations are highly overlapping and barely significantly different. Indeed, other factors may contribute to segregation bias, such as cohesion fatigue and/or kinetochore size as proposed in previous work (Worrall et al, Cell Reports 2018; Drpic et al, Current Biology 2018). Currently, the relative contributions of these different aspects of chromosome biology to missegregation rates remains unclear, and at the minimum needs a more balanced discussion.

Finally, although this paper focuses on establishing the role of chromosome positioning in missegregation rates, there is little new mechanistic insight into how this occurs. Further investigation into the mechanism (for example, do peripheral chromosomes actually experience more merotelic attachments?) would strengthen the authors' arguments and assuage concerns about the correlative nature of the existing data.

Other points

(1) Because it is of critical importance, the monastrol treatment experiment should be performed with at least one other cell line.

(2) For live-tracking dCas9 experiments, the authors should provide more information about how "central" and "peripheral" were defined. From the methods: "Relative nuclear position of tracked chromosomes at the last frame of G2 were determined by dividing the nucleus into 3 equally spaced concentric areas using ImageJ". Were chromosomes in the outside ring "peripheral" and chromosomes in the inside ring "central"? Were some chromosomes in between not counted?

(3) In the methods for the monastrol experiment, the authors state that the center of the cells was "manually estimated". This makes this quantification seem somewhat subjective; therefore, it would be best to have a more rigorous way to define the center.

(4) Some figures are missing statistical significance tests: Fig. 3K; Fig. S3B,E

(5) In Fig. S2B, the number of cells assessed should be reported.

(6) Page 4, line 94: The word "probability" is spelled incorrectly.

Referee #2 (Remarks to the Author):

Klaasen et al. propose that the frequency of missegregation of chromosomes is determined by their position in the interphase nucleus. In support, using single cell sequencing methods, the authors show differences in the missegregation frequency of chromosomes in various cell lines and in organoids with larger chromosomes generally more frequently mis-segregated. The authors also show that mis-segregated chromosomes are more frequently included in micronuclei. Randomization of the location of chromosomes on the metaphase plate and live cell tracking alters missegregation frequency and confirms the authors' proposal.

This is a nicely simple paper that addresses the long-standing question of whether the frequency of missegregation differs amongst chromosomes and if so, what determines these differences. I found the data largely convincing, although most conclusions are based on correlations. The authors' proposal that larger chromosomes which are also found more frequently at the nuclear periphery take longer to align in the metaphase plate and thus have a higher likelihood of missegregation is very plausible, although it is not tested in this study.

The data in figure 1 showing differential missegregation frequencies is not novel and confirms earlier studies, particularly by Worall et al., Cell Reports, 2018 and various less systematic studies.

The novel aspect of this study is the link to interphase position. The data in support of that claim are largely based on correlation and are not quite as strong as one would like. The only direct, although very qualitative, data to show that it is indeed the interphase position of a chromosome that determines its missegregation frequency is the Cas9 experiments in figure 4. The "randomization" experiment using monastrol does not alter interphase position but only the alignment behavior of chromosomes and as such does not directly speak to the role of interphase position.

Additional experiments to strengthen this key point should be included:

It would be reassuring to more directly demonstrate the preferential missegregation and incorporation into micronuclei, for example, by use of one of the cell lines which allow tethering of a genome locus to the lamina (Reddy et al., Nature, 2008; Kumaran et al., JCB, 2008; Finlan et al. PLOS Gen, 2008). The prediction would be that the chromosomes containing the tagged loci should be frequently missegregated and present in micronuclei. |

The interphase position of chromosomes is cell-type specific. One would thus predict that different chromosomes will be missegregated in different cell types. From the authors' limited data, it is not quite clear that this is the case. Can they show clear examples of a chromosome that shows different location in different cell-types and the corresponding missegregation frequencies. Rather than rely on published literature on chromosome location, ideally they would determine the distribution of these chromosomes in interphase nucleus in multiple cell types and show differential segregation behavior in different cell types.

The obvious implication of these findings is that differences in missegregation have consequences for cancer cells. A prediction would be that the most frequent aneuploidies in a given cancer type predict the location of the aneuploid chromosomes. Is that the case?

Referee #3 (Remarks to the Author):

Reviewer's comments

Nuclear position of chromosomes dictates probability of missegregation and micronuclear entrapment

Sjoerd J. Klaasen, My Anh Truong, Richard H. van Jaarsveld, Sippe de Vries, Elianne M. Gerrits, Kim de Luca, Jop Kind, Susanne Lens & Geert J.P.L Kops

Klaasen et al. have elegantly investigated the initial steps in formation of aneuploidy and the link between nuclear chromosome position, segregation error rate and micronuclei entrapment. Their excellent study demonstrates that chromosome location can cause a segregation error bias with peripheral chromosomes taking longer to align and exhibiting elevated error rates compared to central chromosomes. Their findings provide insights into the bias in causes of aneuploidy, and the non-random substrate for selection and opens up several potential questions to further our current understanding of chromosome missegregation and tumour evolution. With revision I believe that this manuscript is suitable for the readership of Nature.

- To further understand the relevance of these findings to tumour evolution, it would be important to compare these segregation error biases between a normal cell of origin and a cancer cell from the same tissue. How recurrent are these changes between cells from different tissue types? This could provide an understanding as to whether there is any change in nuclear positioning of specific chromosomes during malignant transformation and whether these changes impact subsequent mutations. On a similar note, the authors could explore the distribution of essential genes on more central vs more peripheral chromosomes in different contexts and identify whether they are preferentially located on chromosomes that are less likely to be missegregated and lost in their analysis.
- Considering the prevalence of genome duplication in several cancer types and the important role suggested from this current study of nuclear positioning and chromosome segregation, it would be valuable to explore the impact of genome duplication in this setting. Can the authors use their time lapse methodology to explore whether nuclear positioning and error rates are preserved following genome duplication or binucleation?
- Some chromosomes have an overall elevated missegregation rate but relatively low extents of partial losses and gains e.g. chromosome 18 in Figure 1C. Could this reflect a role for additional factors influencing missegregation rate biases. On this note, given the association between replication timing and lamina associated domains, is it possible that this in turn impacts specific missegregations and in part could be responsible for the biases observed. Using publically available data on replication timing, perhaps it may be helpful to disentangle the relative contributions of replication timing, chromosome boundaries with the extent of missegregation/ partial chromosome losses.
- One question that arises from these studies is how conserved are these chromosome locations between different tissues or cancer types and does this impact which chromosomes may ultimately be more frequently lost in certain contexts such as in primary tissues or metastases?
- Can the authors distinguish between the actual time taken to align large chromosomes vs the actual distance travelled by the chromosome as the factor most responsible for influencing the missegregation rate? If nuclear envelope breakdown to metaphase time is prolonged, would a small central chromosomes have an equivalent missegregation rate to large peripheral chromosome.
- In figure 3C. it is not clear which comparisons are being made for the quantifications. Is the distance of chromosome 1 being compared to that of chromosome 15, 2 vs 17 and 6 vs 18? If so, why are these specific combinations compared?

- The data from Figure 4 suggest that misalignment of chromosome pairs itself could be a determining factor of later segregation errors. So, the distance between chromosome pairs could be a determining factor of segregation error. This begs the following questions: 1. is there a difference in segregation error rate between chromosome pairs that have large inter-pair distances relative to small inter-pair distances? 2. Do chromosome pairs of which both chromosomes are located in the centre or periphery, have greater segregation error rates depending on the inter-pair distance?

Author Rebuttals to Initial Comments:

Rebuttal to the referee comments:

Referee #1 (Remarks to the Author):

Major points

(1) In previous papers (Worrall et al, Cell Reports 2018; Drpic et al, Current Biology 2018), monastrol treatment, led to an increase in biased segregation, not the decrease reported here. In the case of the Worrall et al paper, they used of the same tissue culture cells studied in the current manuscript. Perhaps this could be explained by the use of monastrol followed by MPS1 inhibition (this study), as opposed to monastrol alone (Worrall). However, this is an important point that needs to be clarified experimentally. The authors should determine if monastrol washout alone still causes biased missegregation, despite randomizing of chromosome position. If this were to be the case, absent another explanation, this finding would contradict the authors' model.

It is indeed relevant to compare our approach with the one of the Worrall et al study, who use similar cells. The difference with our approach is that they used a 8h monastrol treatment, which, similar to their 8h nocodazole treatment-and-release experiment, resulted in cohesion fatigue of predominantly chromosomes 1 and 2. In our experiments we used a 4h monastrol treatment, followed by release in the presence of Cpd-5 to induce missegregations. As we show in Extended Data Fig. 7b, a 4h treatment did not result in substantial missegregations and therefore likely has not yet resulted in cohesion fatigue. We thus conclude that the missegregations induced by Cpd-5 indeed were random, and not masking a monastrol-release-dependent bias.

(2) Although there is a correlation between interphase chromosome position and missegregation frequency, it is not perfect. This reviewer wonders if the correlation could be strengthened if the author looked directly at chromosome position in mitosis and correlated it with missegregation rates (e.g., using H2B-Dendra2).

We updated one and now include two additional such experiments. In the first, we tracked centromeres by expressing CENPA-GFP, and show that peripheral chromosomes take longer to align and more frequently missegregated than centrally located ones (Fig. 3e-h). In the second experiment, we live tracked chromosomes 1 and 9 by Cas9-labeling, and show that if a chromosome was close to the periphery at the onset of mitosis, regardless of its identity, it was more likely to missegregate (Fig. 4d-f). In the third, similar, experiment we live tracked chromosome 11 with an integrated LacO array by LacI-GFP labeling, and show that it is more likely to missegregate when close to the periphery (Extended Data Fig. 8c,d).

(3) The strongest direct experimental evidence for position causing missegregation comes from the monastrol washout experiment. But it remains possible that this global disruption of spindle function might introduce its own set of segregation biases. It would improve the manuscript if it were possible to alter chromosome position without globally disrupting spindle assembly. Perhaps the authors could adapt engineered systems where individual chromosomes can be tethered to the nuclear periphery (Finlan et al PLoS Genetics 2008).

We thank the reviewer for this excellent suggestion. We have generated a cell line in which the predominantly internally located chromosome 9 can be experimentally repositioned to the nuclear

periphery by virtue of Cas9 and the rapalog conditional dimerisation system. In line with what is predicted from our hypothesis, a two-fold increase in the number of cells with chromosome 9 at or near the nuclear periphery after induced dimerisation with Lamin B1 caused a two-fold increase in segregation errors of chromosome 9 (fig. 4g-i).

(4) In addition to the monastrol experiment, the authors employ several methods to assess the correlation between interphase chromosome position and missegregation rates. However, chromosome position does not, in fact, fully explain error rates: this point should be better emphasized in the text. For example, aneuploidy rates in RPE-1 cells actually correlate better with chromosome size (Fig. S7C, $r = 0.8201$) than with LAD frequency (Fig. 3A, $r = 0.7611$) or chromosome position (Fig. S7F, $r = 0.7790$). Additionally, in Fig. 3H, the distributions of the “distance to center” measurement for the “Error” vs. “No error” populations are highly overlapping and barely significantly different. Indeed, other factors may contribute to segregation bias, such as cohesion fatigue and/or kinetochore size as proposed in previous work (Worrall et al, Cell Reports 2018; Drpic et al, Current Biology 2018). Currently, the relative contributions of these different aspects of chromosome biology to missegregation rates remains unclear, and at the minimum needs a more balanced discussion.

With our direct chromosome tracking and conditional repositioning experiments (fig. 4, and Extended Data fig. 8), we hope to have convinced the reviewer that in conditions where all other variables are equal (chromosome size, kinetochore size, cohesin levels, etc) the position of a chromosome directly impacts its chance to missegregate. We fully agree with the reviewer that other biases are likely to contribute also, and we discuss in the final paragraph of the main text our findings in the light of previously reported bias mechanisms.

Finally, although this paper focuses on establishing the role of chromosome positioning in missegregation rates, there is little new mechanistic insight into how this occurs. Further investigation into the mechanism (for example, do peripheral chromosomes actually experience more merotelic attachments?) would strengthen the authors’ arguments and assuage concerns about the correlative nature of the existing data.

This is an interesting point that spurred us to dive deeper into live centromere tracking experiments.

We now show that peripheral chromosomes were on average slower to congress to the metaphase plate, and that this was even more prominent for those that were close to a spindle pole (fig. 3e,f). These chromosomes then were also more likely to missegregate, including as laggards (fig. 3g,h). We observed this in RPE1 cells treated with Cpd-5 as well as in a CIN cancer cell line (fig. 3i-k). A plausible hypothesis is therefore that peripheral chromosomes experience challenges that delay their congression and biorientation, resulting in misalignments and merotelic attachments at anaphase, and that this is exacerbated when they are close to a spindle pole. Follow-up studies are underway to examine the molecular causes of pole-proximity-dependent challenges for biorientation.

Other points

(1) Because it is of critical importance, the monastrol treatment experiment should be performed with at least one other cell line.

We resorted to the (non-CIN) cancer cell line HCT116. scKaryo-seq showed that the non-random segregation errors upon treatment with Cpd-5 were partially randomised when cells were released from 4h monastrol. The data are depicted in Extended Data Fig. 7d-e.

(2) For live-tracking dCas9 experiments, the authors should provide more information about how “central” and “peripheral” were defined. From the methods: “Relative nuclear position of tracked chromosomes at the last frame of G2 were determined by dividing the nucleus into 3 equally spaced concentric areas using ImageJ”. Were chromosomes in the outside ring “peripheral” and chromosomes in the inside ring “central”? Were some chromosomes in between not counted?

We have now included the following information: "chromosomes were considered central if they resided in the innermost two shells or were touching the second most outer ring." (page 30).

(3) In the methods for the monastrol experiment, the authors state that the center of the cells was “manually estimated”. This makes this quantification seem somewhat subjective; therefore, it would be best to have a more rigorous way to define the center.

We have now determined the center of cells using an ImageJ macro which estimates the center based on the DAPI staining. See methods, page 28, for more details.

(4) Some figures are missing statistical significance tests: Fig. 3K; Fig. S3B,E

We apologise for this and have now added the significance assessments.

(5) In Fig. S2B, the number of cells assessed should be reported.

We have added this information.

(6) Page 4, line 94: The word “probability” is spelled incorrectly.

We have corrected this mistake.

Referee #2 (Remarks to the Author):

It would be reassuring to more directly demonstrate the preferential missegregation and incorporation into micronuclei, for example, by use of one of the cell lines which allow tethering of a genome locus to the lamina (Reddy et al., Nature, 2008; Kumaran et al., JCB, 2008; Finlan et al. PLOS Gen, 2008). The prediction would be that the chromosomes containing the tagged loci should be frequently missegregated and present in micronuclei.

We thank the reviewer for this excellent suggestion. We have generated a cell line in which the predominantly internally located chromosome 9 can be experimentally repositioned to the nuclear periphery by virtue of Cas9 and the rapalog conditional dimerisation system. In line with what is predicted from our hypothesis, a two-fold increase in the number of cells with chromosome 9 at or near the nuclear periphery after induced dimerisation with Lamin B1 causes a two-fold increase in segregation errors of chromosome 9 (fig. 4g-i).

The interphase position of chromosomes is cell-type specific. One would thus predict that different chromosomes will be missegregated in different cell types. From the authors' limited data, it is not quite clear that this is the case. Can they show clear examples of a chromosome that shows different location in different cell-types and the corresponding missegregation frequencies. Rather than rely on published literature on chromosome location, ideally they would determine the distribution of these chromosomes in interphase nucleus in multiple cell types and show differential segregation behavior in different cell types.

This is an interesting point that could provide additional correlative evidence for our hypothesis, as well as touch upon the question whether cell type differences in chromosome locations can impact aneuploidy landscapes. In the mouse, FISH-based experiments of a sub set of chromosomes have shown some differences in chromosomal locations between tissues (e.g. PMID 15239829). Our centromere FISH and DamID data from human cells, however, show quite striking conservation of chromosome positions in cell lines of different tissue origins (Extended Data Fig. 5g-m, 6a-e). In addition, although human lymphoblasts were reported to have a more peripheral position of chromosome 8 than fibroblasts (PMID 11159939), we could unfortunately not replicate this observation (see figure below). So we have not yet found dividing human cell populations with different chromosome locations to perform our missegregation/scKaryo-seq approach on. We are, however, interested in this question, and are designing a follow-up study to assess chromosome locations in organoid cultures of various human tissues (lung, liver, colon, small intestine, breast, pancreas, skin).

Figure 1: DamID and FISH comparing chromosome organization of lymphoblast and RPE1-hTERT cells.

The obvious implication of these findings is that differences in missegregation have consequences for cancer cells. A prediction would be that the most frequent aneuploidies in a given cancer type predict the location of the aneuploid chromosomes. Is that the case?

We indeed hypothesize that non-random segregation errors impact cancer cells. Since aneuploidy patterns in cancer are correlated with expression profiles of tissue stem cells, we expect the patterns observed by sequencing of late-stage cancer to be predominantly the result of selective pressures during tumor development. The impact of segregation error bias may thus have its most prominent impact on

the evolutionary trajectories and timing of acquisition of specific aneuploidies, possibly further shaped by tissue-specific differences in nuclear chromosome locations. We are very keen on studying this, but it will be a long-term follow-up project because it requires tracking aneuploidy landscapes over time from the moment chromosomal instability manifests, preferably in cells from different tissue origin. In parallel, we are collaborating with computational modeling experts to model dynamics and clonal spreading of relatively simple aneuploidy patterns found in cancer under conditions of yes/no biased missegregation.

Referee #3 (Remarks to the Author):

- To further understand the relevance of these findings to tumour evolution, it would be important to compare these segregation error biases between a normal cell of origin and a cancer cell from the same tissue. How recurrent are these changes between cells from different tissue types? This could provide an understanding as to whether there is any change in nuclear positioning of specific chromosomes during malignant transformation and whether these changes impact subsequent mutations. On a similar note, the authors could explore the distribution of essential genes on more central vs more peripheral chromosomes in different contexts and identify whether they are preferentially located on chromosomes that are less likely to be missegregated and lost in their analysis.

- One question that arises from these studies is how conserved are these chromosome locations between different tissues or cancer types and does this impact which chromosomes may ultimately be more frequently lost in certain contexts such as in primary tissues or metastases?

These are very interesting points. The overall location of chromosomes is quite well conserved between tissues, as evidenced by our centromere FISH and DamID data from human cell lines of different tissue origins (Extended Data fig. 5g-m, S6a-e). However, mouse cells of different origins display different locations of some chromosomes (e.g. PMID 15239829), and human lymphoblasts are reported to position chromosome 8 more peripherally than human fibroblasts (PMID 11159939). To study potential differences in human context, we are designing a follow-up study to assess chromosome locations in organoids cultures of various human tissues (lung, liver, colon, small intestine, breast, pancreas, skin) and measure their aneuploidy landscapes after induced segregation errors. This will also allow us to compare these landscapes to those of transformed tissue-matched tumor organoids, to which we have access.

On a more general note, we indeed hypothesize that non-random segregation errors impact cancer cells. Since aneuploidy patterns in cancer are correlated with expression profiles of tissue stem cells, we expect the patterns observed by sequencing of late-stage cancer to be predominantly the result of selective pressures during tumor development. The impact of segregation error bias may thus have its most prominent impact on the evolutionary trajectories and timing of acquisition of specific aneuploidies, possibly further shaped by tissue-specific differences in nuclear chromosome locations. We are very keen on studying this, but it will be a long-term follow-up project because it requires tracking aneuploidy landscapes over time from the moment chromosomal instability manifests, preferably in cells with different tissue origin. In parallel, we are collaborating with computational modeling experts to model dynamics and clonal spreading of relatively simple aneuploidy patterns found in cancer under conditions of yes/no biased missegregation.

- Considering the prevalence of genome duplication in several cancer types and the important role suggested from this current study of nuclear positioning and chromosome segregation, it would be valuable to explore the impact of genome duplication in this setting. Can the authors use their time lapse methodology to explore whether nuclear positioning and error rates are preserved following genome

duplication or binucleation?

This is an excellent suggestion, and we have now included single cell sequencing of pseudo-tetraploid human RPE1 cells after one round of induced segregation errors (Extended Data fig. 3e,f). We found that the missegregation bias is similar to that of its diploid counterpart.

- Some chromosomes have an overall elevated missegregation rate but relatively low extents of partial losses and gains e.g. chromosome 18 in Figure 1C. Could this reflect a role for additional factors influencing missegregation rate biases. On this note, given the association between replication timing and lamina associated domains, is it possible that this in turn impacts specific missegregations and in part could be responsible for the biases observed. Using publically available data on replication timing, perhaps it may be helpful to disentangle the relative contributions of replication timing, chromosome boundaries with the extent of missegregation/ partial chromosome losses.

This is an interesting thought, which we had not yet considered. Indeed replication timing could play a role, as could, for example, differences between chromosomes in obtaining breaks when lagging behind and getting trapped in the cytokinetic furrow (PMID 21960636). Regarding chromosome 18: it has a relatively high percentage of LADs and should therefore have more late-replicating regions. We thus expect these regions to persist as DNA catenanes in mitosis and lead to missegregations in our protocol. Yet, that is not what we see in the scKaryo-seq data for chromosome 18. To look at this more broadly, as per the reviewer's suggestion, we used publicly available replication timing data from the Replication Domain Genome Browser, and found a strong correlation between chromosome size or structural variations and the number of late-replicating regions in RPE1-hTERT cells. It is therefore a distinct possibility that late-replicating regions can persist more frequently on chromosomes we identify as having high probability of missegregation, and cause breaks during mitosis.

- Can the authors distinguish between the actual time taken to align large chromosomes vs the actual distance travelled by the chromosome as the factor most responsible for influencing the missegregation rate?

This is an interesting point that spurred us to dive deeper into live centromere tracking experiments.

We now show that peripheral chromosomes were on average slower to congress to the metaphase plate, and that this was even more prominent for those that were close to a spindle pole (fig. 3e,f). These chromosomes then were also more likely to missegregate, including as laggards (fig. 3g,h). We observed this in RPE1 cells treated with Cpd-5 as well as in a CIN cancer cell line (fig. 3i-k). On the other hand, central chromosomes that are relatively far from the metaphase plate biorient much faster, suggesting it is not about the distance to the metaphase plate, but about the actual time taken to align (Extended Data Fig. 6f-k). A plausible hypothesis is therefore that peripheral chromosomes experience challenges that delay their congression and biorientation, resulting in misalignments and merotelic attachments at anaphase, and that this is exacerbated when they behind the spindle pole. Follow-up studies are underway to examine the molecular causes of pole-proximity-dependent challenges for biorientation.

If nuclear envelope breakdown to metaphase time is prolonged, would a small central chromosomes have an equivalent missegregation rate to large peripheral chromosome.

In essence, this is what the monastrol release experiment in figure 4a-c has assessed. The answer is that

indeed prolonging the time from NEB to metaphase equalizes the probability of missegregation to random level.

- In figure 3C. it is not clear which comparisons are being made for the quantifications. Is the distance of chromosome 1 being compared to that of chromosome 15, 2 vs 17 and 6 vs 18? If so, why are these specific combinations compared?

Indeed, we assessed combinations of large and small chromosomes. In combining FISH probes, we are limited to only a few per experiment (for imaging reasons), and as such we reasoned that combining differently sized chromosomes provided the best experimental condition with the optimal, internal controls for experimental variations.

- The data from Figure4 suggest that misalignment of chromosome pairs itself could be a determining factor of later segregation errors. So, the distance between chromosome pairs could be a determining factor of segregation error. This begs the following questions: 1. is there a difference in segregation error rate between chromosome pairs that have large inter-pair distances relative to small inter-pair distances? 2. Do chromosome pairs of which both chromosomes are located in the centre or periphery, have greater segregation error rates depending on the inter-pair distance?

We expect that homologous chromosome pairs behave independently from each other in mitosis. The distance between pairs is therefore unlikely to be a factor in determining the eventual probability of missegregation. We nonetheless looked into whether our data can address this question. In general, because of geometrical considerations, peripheral chromosome pairs (homologs) have a higher chance of being far apart than central ones do. Indeed, chromosome pairs that are further apart from each other have a higher chance to missegregate. We were not able to answer the second point, because the number of double peripheral or double central homologs was too low to draw any meaningful conclusions.

Reviewer Reports on the First Revision:

Referees' comments:

Referee #1 (Remarks to the Author):

The authors have done an excellent job of thoroughly addressing my comments with new experiments and qualified discussion. I support publication of this paper without further revision.

Referee #2 (Remarks to the Author):

The revised manuscript by Klaasen et al. addresses most of the points made in response to the first submission. The newly added experiment of tethering chromosome 9 to the periphery is convincing and very strongly supports the authors' central claim of location being an important determinant of segregation defects. These are novel and important findings.

Referee #3 (Remarks to the Author):

Klassen et al have suitably revised the manuscript and it's clear that most of the comments have been addressed adequately.

It's also great to hear that the authors are interested in tracking aneuploidy landscapes over time, which is likely to significantly contribute to our understanding of aneuploidy and cancer.

The question I was hoping the authors would address at this stage was simply the difference between nuclear lamina positions between normal cells and cancer cells e.g. pneumocytes vs a LUAD cell line to assess possible differences in chromosome locations between a cell-of-origin vs cancer cell as opposed to the positional similarities between cancer cells from different tissues.

Subsequently, one could leverage published data, and correlate the load of cancer associated mutations and essential genes etc on different chromosomes in relation to whether these chromosomes frequently undergo relocalisation during malignant transformation in a given cancer type.